# Drivers of high frequency extreme sea level around Northern Europe - Synergies between recurrent neural networks and Random Forest

Céline Heuzé[1], Linn Carlstedt[1,2], Lea Poropat[3], and Heather Reese[1]

[1]Department of Earth Sciences, University of Gothenburg, Gothenburg, Sweden
[2]Department of Research and Development, Swedish Meteorological and Hydrological Institute, Gothenburg, Sweden
[3]National Centre for Climate Research, Danish Meteorological Institute, Copenhagen, Denmark

**Correspondence:** Céline Heuzé (celine.heuze@gu.se)

**Abstract.** Northern Europe is particularly vulnerable to extreme sea level events as most of its large population, financial and logistics centres are located by the coastline. Policy makers need information to plan for near- and longer-term events. There is a consensus that for Europe, in response to climate change, changes to extreme sea level will be caused by mean sea level rise rather than changes in its drivers, meaning that determining current drivers will aid such planning. Here we determine from explainable AI the meteorological and hydrological drivers of high frequency extreme sea level at nine locations on the wider North Sea - Baltic coast using Long Short Term Memory (LSTM, a type of deep recurrent neural network) and the simpler Random Forest regression on hourly tide gauge data. LSTM is optimised for targeting the excess values, or periods of prolonged high sea level; Random Forest, the block maxima, or most extreme peaks in sea level. Through permutation feature of the LSTM, we show that the most important driver of the periods of high sea level over the region is the westerly winds, whereas the Random Forest reveals that the driver of the most extreme peaks depends on the geometry of the local coastline. LSTM is most accurate overall, although predicting the highest values without overfitting the model remains challenging. Despite being less accurate, Random Forest agrees well with the LSTM findings, making it suitable for predictions of extreme sea level events at locations with short and/or patchy tide gauge observations.

## 1 Introduction

About 50 million people live by the coast in Europe (Neumann et al., 2015). In northern Europe in particular, many strategically crucial financial and logistical hubs are also located by the coast, making population and infrastructure vulnerable to extreme sea level events (see review by van de Wal et al., 2024, and references therein). As the global climate warms and sea level rises, extreme sea level events are projected to increase both in magnitude and frequency, especially so on the North Sea and Baltic coasts (Vousdoukas et al., 2017). As recently reviewed by Melet et al. (2024), there is a consensus that this increase is driven by the shifting baseline of sea level rise, rather than by changes in the mechanisms driving the extreme events. This means that identifying these drivers now would aid policy makers in better planning of near- (van den Hurk et al., 2022) and longer-term (Groeskamp and Kjellsson, 2020) needs for coastal defences.

The drivers of sea level in Northern Europe have been extensively studied with hydrodynamic modelling (see review in Melet et al., 2024). These models showed that from seasonal to multi-decadal scales, variability is primarily controlled by the

atmosphere, especially westerly winds (e.g. Frederikse and Gerkema, 2018; Tinker et al., 2020). Due to the models' coarse

resolution, especially for the atmosphere, data-driven approaches are more adapted for shorter time scales. Using daily altimetry data over a small region of the North Sea, Sterlini et al. (2016) found a similar relationship between sea level and zonal winds, but also that the wind component most important for sea level is strongly dependent on the coast's geometry. Sterlini et al. (2017) expanded their region of study to the entire North Sea, and found likewise that for daily sea level, the location

influenced whether meteorological or steric components mattered most. From hourly tide gauge data, Marcos and Woodworth (2017) found the same strong relationship between the steric component and extreme sea level values (but did not investigate any possible relationship with atmospheric variables).

Globally, some tide gauge records date back to the mid 1800s (Haigh et al., 2023). This data richness means that data-driven approaches involving machine learning are an obvious choice for sea level research. Most often, these methods aim to forecast

the non-tidal residuals, using atmospheric parameters as predictors. For example, Ishida et al. (2020) reproduced hourly tide gauge data in Osaka, Japan, using a type of recurrent neural network called Long Short Term Memory (LSTM, see section 2.4) and reanalysis-based time series of wind speed, wind direction, sea level pressure, and air temperature, as well as the global air temperature as a remote global warming forcing. Hieronymus et al. (2019) used a similar approach for 9 tide gauge stations on the Swedish coast, except that they used the full spatial fields of the reanalysis variables instead of time series, and showed that

the 36h forecasts they generated were faster and more accurate than those by the best European hydrodynamic model. Using the HIDRA2 (Rus et al., 2023) encoder-based deep network to forecast sea level at 5 tide gauge stations along the Estonian coast, Barzandeh et al. (2024) found the same result: machine learning methods produce better forecasts, faster, than state-of-the-art hydrodynamics models. They do note that the network struggles to reproduce high-frequency variability, producing too smooth timeseries, a result that Tadesse et al. (2020) too found for daily sea level, globally, using Random Forest.

Predicting extreme values is not a problem unique to sea level, and therefore the development of machine-learning based methods adapted to extreme values is ongoing for many climate applications. One main direction is to use convolutional neural networks on spatial fields, for example for predicting extreme winds (Jiang et al., 2022a), precipitation (Wilson et al., 2022), sudden stratospheric warming events (Strahan et al., 2023), or tropical cyclones (Ascenso et al., 2024). These topics benefit from the fact that the researchers can somewhat easily augment their data by rotating their images, hence generating new

training points (Ascenso et al., 2024). This cannot be done for 1D time series analysis, which instead preferably uses LSTM. Recent examples of this include predicting European river flooding (Jiang et al., 2022b), storm intensity on the French Atlantic coast (Frifra et al., 2024), or extreme precipitation at specific locations in China (Tang et al., 2022). Note that Tang et al. (2022) also uses Random Forest.

What all these studies have in common is that their main objective is to predict extremes, rather than identify what drives

them, even though both LSTM and Random Forest can be made explainable. They also often rely on networks that are overly fitted to a specific location, and therefore of limited use to a wider region. Here we develop LSTM and Random Forest models to predict and identify the drivers of sea level around the wider North Sea and Baltic regions, using hourly tide gauge data as predictand and atmospheric and hydrological time series as predictors, focussing on extreme sea level only, as we describe in

section 2. We present the results of the LSTM and Random Forest-based analyses in sections 3.1 and 3.2, respectively, before discussing their applicability to sea level monitoring and coastal defense planning in section 3.3.

## 2 Methods

### 2.1 Hydrographic, meteorological, and hydrological data

We use hourly tide gauge data from nine stations around Northern Europe (Fig. 1a): three on the North Sea coast (station names Lowestoft, Den Helder, and Esbjerg), three in the Baltic (Ratan / Umeå, hereafter referred to as Umeå, Helsinki, and Gedser), and three in the transition between the two seas, the Skagerrak / Kattegat (Oslo, Gothenburg / Göteborg, and Klagshman / Malmö, hereafter referred to as Göteborg and Malmö, respectively). For each region, we tried to select locations that are major population centres but had to compromise to obtain long enough, uninterrupted time series (Table 1). We also excluded cities where the water level is artificially controlled by humans, such as London or Rotterdam that have flood defense systems, or Stockholm that is fed by a reservoir lake. For the three Swedish cities Göteborg, Malmö and Umeå, the tide gauge data were provided by the Swedish Meteorological and Hydrological Institute; for the other locations, we used the Global Extreme Sea Level Analysis (GESLA) dataset version 3, last updated in November 2021 (Woodworth et al., 2016; Haigh et al., 2023).

To determine the potential drivers of extreme sea level, we use meteorological and hydrological data. Ideally, we would have used meteorological time series from weather stations collocated with the tide gauge stations. Unfortunately, those are often at different locations, their distribution managed by different services, and their obtention requiring that one speaks the language of the country to understand the download interface. In addition, the time series are often patchy, missing data at different times than the tide gauge stations. Therefore, we opted to use the ERA5 reanalysis instead (Hersbach et al., 2020). The spatial resolution is 0.25° and the temporal resolution hourly; the time period covered 1 January 1940 to 31 December 2023. We use the hourly 10 m u- and v-components of the wind, evaporation, total precipitation, mean sea level pressure and sea surface temperature. Wind and sea level pressure have a dynamic effect on the sea level; evaporation and precipitation change the amount of water available, and along with sea surface temperature, contribute to steric sea level. We compute the wind speed and direction from the u- and v- components. As we want to determine the feasibility of predicting sea level from in-situ stations, for each city and variable, we use the time series of the ERA5 grid cell closest to the tide gauge station coordinates as provided by SMHI or GESLA, similar to what Ishida et al. (2020) did. We also generate the hourly timeseries of three "remote" drivers, representing the state of the wider North Atlantic climate and its storminess:

– The hourly spatial-minimum sea level pressure around Iceland (Fig. 1b, blue), longitudes -30 to -10°E and latitudes 56 to 66°N;

– The hourly spatial-maximum sea level pressure around the Azores (Fig. 1b, orange), longitudes -37 to -22°E and latitudes 32 to 42°N;

– And the hourly spatial-mean sea surface temperature over the eastern North Atlantic (Fig. 1b, pink), longitudes -40 to -10°E and latitudes 40 to 60°N.

Note that combining the Icelandic low and Azores high time series yields the North Atlantic Oscillation index (e.g. Hurrell, 1995).

The last potential driver of extreme sea level used in this study is hydrological. Data obtention and quality issues are similar to those of the meteorological data described in the previous paragraph. We therefore use the river discharge time series from the Global Runoff Data Centre (GRDC). We manually selected the station(s) of the rivers that discharge in the city; some had none, some had up to three (Table 1). The GRDC data are daily, meaning that we linearly interpolated them to generate hourly data. We acknowledge that daily mean data are not ideal to detect and predict extreme hourly values of sea level, and can only lament that hourly products are not available. Even as daily means, hydrological data often have a shorter time coverage than hydrographic ones (Table 1); for the few locations around Sweden where we found hourly data, their time coverage was too short and patchy to be of use.

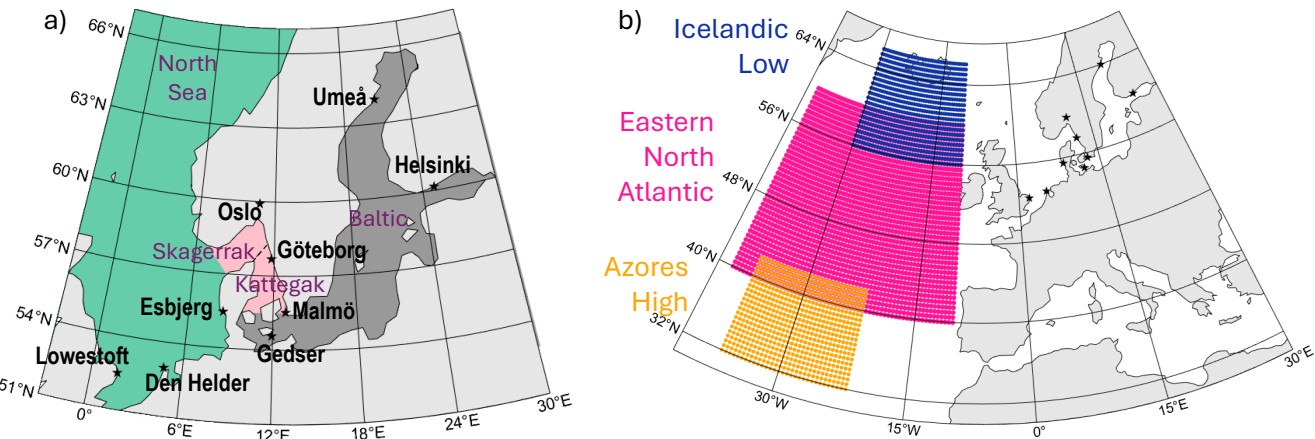

**Figure 1.** a) The nine cities of this study, chosen for their data availability and to cover the three main maritime regions: The North Sea (green), the Skagerrak/Kattegat (salmon), and the Baltic (grey). b) Regions used for the creation of the remote predictors: The Icelandic Low (blue) and Azores High (orange) sea level pressure, and the Eastern North Atlantic (pink) sea surface temperature.

## 2.2 Further data preparation

We de-tided the tide gauge data using the UTide package for Matlab (Codiga, 2024). Although some machine learning methods have used the tide gauge data including their tide signal (e.g. Rus et al., 2023), we chose to remove them as they are the predictible part of the signal, and we are interested in explaining the rest.

All datasets have been de-trended, assuming a linear trend and using a 95% significance threshold. The exceptions are the u- and v- components of the wind, which were first split into their positive (westerly resp. southerly) and negative (easterly resp.

| Station name (Country) | Sea level period | Rivers | GRDC station | Rivers' common period |
|---|---|---|---|---|
| Den Helder (NL) | 1932 - 2017 | N/A | | |
| Esbjerg (DK) | 1950 - 2019 | Kongea | 6934370 | 1934 - 2023 |
| | | Ribe | 6934350 | |
| Gedser (DK) | 1892 - 2012 | N/A | | |
| Gothenburg / Göteborg (SE) | 1968 - 2022 | Säveån | 6233326 | 1979 - 2023 |
| Helsinki (FI) | 1971 - 2019 | Vantaanjoki | 6854115 | 1937 - 2023 |
| Lowestoft (UK) | 1964 - 2020 | Waveney | 6606900 | 1964 - 2020 |
| Klagshamn / Malmö (SE) | 1930 - 2023 | Huje | 6233190 | 1965 - 2023 |
| | | Segeå | 6233367 | |
| Oslo (NO) | 1915 - 2020 | Grytbekken | 6729360 | 1968 - 2018 |
| | | Saternbekken | 6729425 | |
| | | Sandvikselva | 6729420 | |
| Ratan / Umeå (SE) | 1892 - 2023 | Umeälven | 6233501 | 1919 - 2017 |

**Table 1.** Maximum time period for which the hydrographic data are available for each city, and corresponding hydrological data: River names, river station number in the Global Runoff Data Centre (GRDC), and common time period of the rivers if there are more than one. See Methods and Data availability sections.

northerly) parts such that:

$$
\begin{cases}
u_+ = u \text{ if } u > 0 \\
u_+ = 0 \text{ otherwise}
\end{cases}
\tag{1}
$$

and

$$
\begin{cases}
u_- = 0 \text{ if } u > 0 \\
u_- = u \text{ otherwise}
\end{cases}
\tag{2}
$$

Then these positive and negative components were detrended, and used as predictors instead of the u- and v- components. The other exception is the wind direction, which was not detrended. We purposely keep variables that are correlated to each other (see Appendix Tables A1 to A3) to test compound events. We acknowledge that this may result in an underestimation of the importance of the individual predictors. The correlations will be discussed in the Results section when relevant. The predictor short names and their definition are summarised in Table 2.

As we describe in subsection 2.4, we use these data in two types of machine learning models: Long Short Term Memory (LSTM), and Random Forest. Random Forest requires no data normalisation, so the variables are used directly after de-tiding and de-trending. For the LSTM, we use a min-max normalisation so that all variables are between 0 and 1. Prior to

normalisation, we convert all time series to their absolute value; this affects only $u_-$ and $v_-$. This is to preserve their shape as zero-inflated, heavy-tailed distributions after normalisation, similar to that of the other variables.

All datasets are in UTC time; no re-timing is needed. Any missing value in the hydrographic or hydrological series was set to 0. We select only the time period common to all three data sources (Table 1, and ERA5 is 1940-2023). The shortest period is 43 years for Göteborg; the longest is 77 years for Den Helder and Umeå.

| Predictor | Definition |
|---|---|
| evap | Local value of the evaporation |
| msl | Local value of the sea level pressure |
| $msl_{azo}$ | Remote driver, sea level pressure over the Azores High region |
| $msl_{ice}$ | Remote driver, sea level pressure over the Icelandic Low region |
| rivers | Runoff of the local rivers (if any) summed |
| sst | Local value of the sea surface temperature |
| $sst_{ENA}$ | Remote driver, sea surface temperature over the Eastern North Atlantic region |
| tprecip | Local value of the total precipitation |
| $u_-$ | Local value of the u component of the wind, negative/easterly values only |
| $u_+$ | Local value of the u component of the wind, positive/westerly values only |
| $v_-$ | Local value of the v component of the wind, negative/northerly values only |
| $v_+$ | Local value of the v component of the wind, positive/southerly values only |
| wdir | Local value of the wind direction |
| wspeed | Local value of the wind speed |

**Table 2.** Summary of the fourteen predictors used in this study, by alphabetical order of the short names used on the figures. See Fig. 1 for the region definitions and Table 1 for the rivers.

## 2.3 Extreme sea level events - definitions

Two types of events are investigated here:

- Peaks in sea level, or absolute maxima of a given block, for which Random Forest is most suited (see Introduction);

- Prolonged periods of high sea level, for which LSTM is most suited.

For the first type of event, we select all values in excess of the mean sea level plus three standard deviations (or above 0.75 on the normalised series used for illustration, Fig. 2). We use a block size of 7 days, i.e. if more than one value is above the threshold within a 7-day period, we select only the maximum value. For all cities, we obtain 200 to 300 events during the time period common to all three types of variables (black stars on Fig. 2).

For the second type of event, we first compute the 30-day running mean time series, and then select the values in that running mean series that are above its mean plus three standard deviations. The value of 30 days was chosen as a compromise: it yields

enough points for training; it is long enough compared to our hourly data, according to extreme value theory (Ólafsdóttir, 2024); and it is longer than the memory of the system (2-3 days, Hieronymus et al., 2019). For all cities, we obtain several thousand hourly data points during the time period common to all three types of variables (red crosses on Fig. 2).

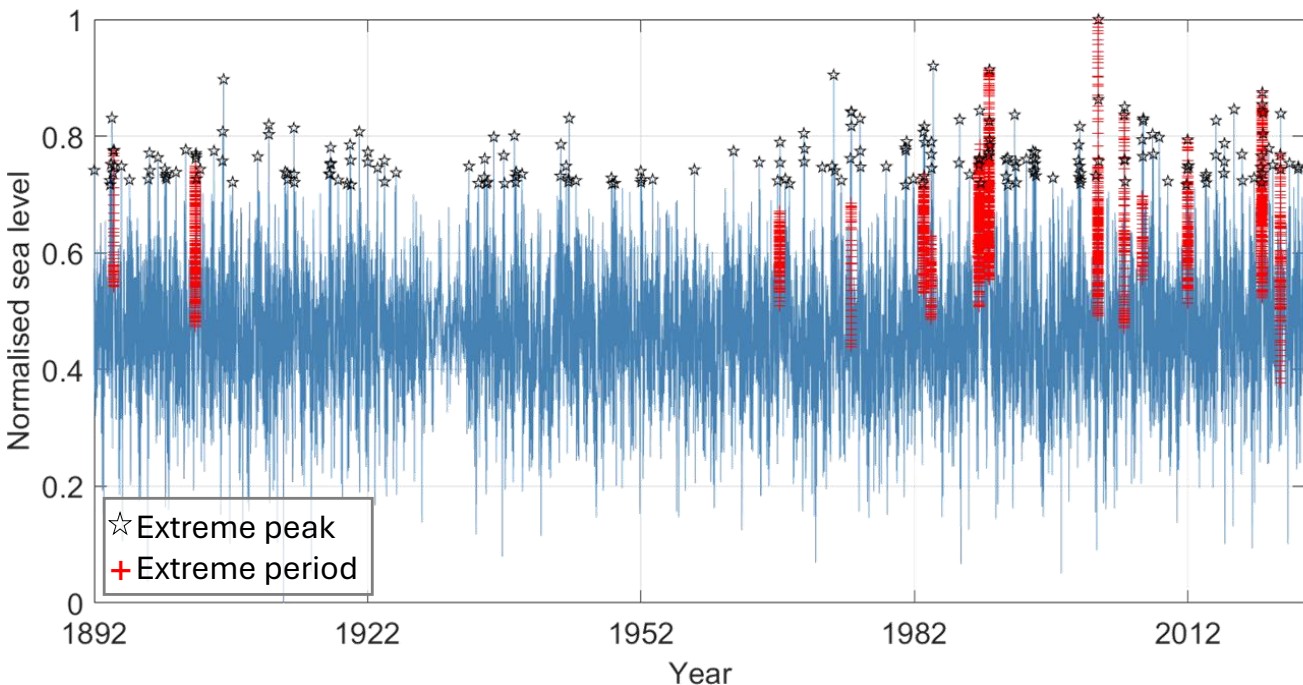

**Figure 2.** Illustration of the two extreme sea level detection methods, using the complete time series of sea level for Umeå, min-max normalised after de-tiding and de-trending. The 212 black stars are the hourly values detected as peaks, and the 5828 red crosses the hourly value detected as belonging to a period of high sea level. The actual detection is done on the shortened time series, and on the non-normalised series for the peaks, see text.

## 2.4   Machine learning models: Long Short Term Memory (LSTM) and Random Forest

### 2.4.1   LSTM and permutation feature importance

Sea level is the result of current and past cumulative forcings. To take this temporal dependency into account, we use a type of recurrent neural network (RNN) called Long Short Term Memory (Hochreiter, 1997). A standard RNN uses as input not only the predictors at the same time step as the target, but also its hidden state variables of the previous time step. Each time the network moves forward, the hidden state is overwritten. LSTM in contrast uses a set of "gates" to regulate which information is forgotten and which is stored and passed as input, allowing the network to build a sophisticated combination of all the past time steps that it considers relevant.

We split the de-trended, normalised time series merging the prolonged periods of high sea level into a validation set (first
15%), a test set (second 15%) and a training set (last 70%). We performed a hyperparameter search on the Göteborg time series,
using the following hyperparameter space:

- window size: 2, 3, 6, 12, 18, 24, 36, 48, 72, 120, 168h;

- number of layers: 1, 2, 3, 4, 5;

- number of units per layer: 10, 20, 33, 50, 100, 200;

- learning rate: 0.001, 0.005, 0.01;

- batch size: 20, 50, 100;

- dropout rate: 0.005, 0.01, 0.015.

After this hyperparameter search, we settled for a network with 3 LSTM layers, with a hyperbolic tangent activation function,
of 100 units each and one dense layer, with a drop out rate of 0.015 in between each layer. We used the Python package Keras
(Chollet and The Keras Team, 2015). The input batch size is 20, and the network's overall learning rate is 0.01. We found the
best performance with a window of 12h, except for Helsinki (24h) and Malmö (36h). These short windows were expected given
that extreme sea level is a short-lived event, and are consistent with the findings of Hieronymus et al. (2019) for the Swedish
coast. The performance for Helsinki was markedly improved by using 10 units per layer and a batch size of 100. These settings
minimised the mean square error, while resulting in a correlation between the target and the predicted value of at least 0.81, i.e.
explaining more than 2/3 of the signal. For each city we then generated 100 models with a random initial state, and selected
the best model following these steps:

- The models are of equally good mean square error by design. We prioritise those that successfully explain most of the
  signal and therefore select the subset of models with a correlation within 0.2 of the maximum correlation of all 100
  models;

- Within this subset, we rank the models based on their overall root mean square error (RMSE) but also their RMSE for
  normalised sea level values larger than 0.66, as we noticed during training that the LSTM struggled to reproduce these
  high values without overestimating the rest of the series;

- We select the model with the lowest sum of these two ranks; at equal value, we take the one with the highest correlation.

To determine the contribution of each predictor to the overall prediction, we perform a permutation feature importance.
That is, for each predictor, we run the model again after having set this predictor to an array of random values. The difference
between the explained variance (= correlation squared) of the default prediction and that with the random values directly gives
the contribution of that predictor to the signal's variance.

### 2.4.2 Random Forest and feature importance

To investigate the peaks in more details, we move away from neural networks and use instead a method that is simpler but more adapted to point measurements: a Random Forest regression. A Random Forest is an ensemble of decision or regression trees, which makes it a preferable method to identify the drivers of sea level as the trees directly choose the most relevant predictors at each split. The Random Forest feature importance is calculated using the Gini importance measure. An inconvenient aspect is that the feature importance returned by the trees is relative to the parameters used in the model, unlike that of the LSTM that is absolute. Another limitation of Random Forest is that temporal relationships are not considered, even though we know that sea level does not depend only on synchronous forcings. We remedy this by providing as input the predictors (listed in Table 2) at the same hourly time step as the target sea level values, but also 1h, 3h, 6h, 12h, 24h, 48h, 3 days, 5 days, and 7 days prior, resulting in a total of 140 predictors.

We randomly split the values into a training set (80%) and a test set (20%). Here too we performed a hyperparameter search on the Göteborg series, with the following hyperparameter space:

- number of trees: 100, 200, 500, 1000, 2000;

- number of splits: 1, 2, 3, 4;

- number of leaves: 2, 3, 4, 5;

- bootstrapping: true, false;

- if bootstrapping true, maximum amount of samples: 0.1, 0.2 ,0.5, 0.75, 1.

After this hyperparameter search, we chose the settings that minimised the squared error, which was 200 trees with a minimum of 2 leaves (min samples leaf) and 4 splits (min samples split), with bootstrap or "bagging" set to true and the maximum amount of samples set to 0.5 (max samples). The optimum settings and results were the same when minimising a weighted mean error to favour the most extreme values instead. We used the Python package Scikit-learn (Pedregosa et al., 2011). To determine the contribution of each predictor to the overall prediction, we used the built-in regression feature estimator, limiting the selection to 35 features (i.e. a quarter of all possible). For each city we produced an ensemble of 100 models with feature estimation and analyse their mean results.

## 3 Results and Discussion

### 3.1 Northwesterly winds contribute most to periods of prolonged high sea level

Starting with the LSTM applied to periods of high sea level, the correlation between the test dataset and its predicted values is at 0.8 or higher for all cities (correlation squared in Appendix Tables A4). The root mean square error between test and prediction is around 5% for all locations except Esbjerg (9%, Appendix Table A5). Likewise, the root mean square error for

only the highest third of the values is around 2.5% for all locations except Esbjerg (4%, Appendix Table A6). We could not see an obvious reason for Esbjerg's difference, and the RMSE there remains acceptably low, so we do not investigate this further.

The feature permutation from the LSTM directly returns the absolute contribution of each predictor to the variance of the signal. Unsurprisingly, for all cities and all predictors, the explained variance is less if a predictor is set to random values instead of its series (Fig. 3 bottom left triangles, and Appendix Table A4). On average the northerly and westerly winds $v_-$ and $u_+$ are first- and second- most important predictors to the majority of the cities, with $v_-$ contributing to more than 50% of the signal in Gedser, Helsinki and Lowestoft (dark red on Fig. 3). Interestingly, these wind values themselves are not extreme over the periods used by the LSTM (Table 3): For all cities, the median of their normalised values is low to very low, but the series also includes normalised values larger than 0.9. That is, contrary to expectations, the westerly and northerly winds are neither anomalously weak or strong during periods of prolonged high sea level. This demonstrates the strength of the LSTM, capable of detecting patterns beyond usual human statistics.

Most of the predictors are important for some cities but negligible in others. The local evaporation for example contributes to 38% of the variance in sea level for Gedser, but 10% and 1% in Helsinki and Umeå, respectively, the other two cities in the Baltic, and 0% in Malmö, its nearest neighbour. Besides, in all these cases the evaporation is only weakly or not at all correlated to the other predictors (Appendix Tables A1 to A3), i.e. its signal is not diffused within the other predictors. Overall, the importance of a predictor is not similar for cities in the same sea, and the importance of the remote predictors does not increase with proximity to their source (e.g. $msl_{ice}$ contributes 5% to Umeå and 2% to Den Helder).

For some predictors, their contribution is negligible regardless of the city, lower than 10%. This is the case for example of $msl_{azo}$ or the southerly winds $v_+$, which is not surprising given that these variables are rather indicative of good weather conditions over the region; or the rivers with the exception of Oslo (19%), which may be because of their original daily resolution. These variables are not strongly correlated to any of the strong predictors either (Appendix Tables A1 to A3). Similarly, the strong contribution of $sst_{ENA}$ to Umeå and nowhere else, along with the low correlation of $sst_{ENA}$ to anything except the local sst, suggests rather that we are missing a potential driver for Umeå that could be correlated to $sst_{ENA}$, such as the sea ice concentration or thickness. Unfortunately for weather observations, the simpler-to-monitor compound variables (msl, wdir and wspeed) do not contribute significantly more than the individual predictors.

Using the root mean square error of the entire prediction set yields broadly the same results as using the explained variance (Appendix Table A5). Using that of the highest sea level values $RMSE_{0.66}$ in contrast gives surprising results: Some predictions are improved when a predictor is replaced by random values (blue top-right triangles on Fig. 3, and Appendix Table A6). In most cases, this predictor did not contribute much to the explained variance anyway, such as the evaporation for Den Helder or Esbjerg (contribution to the variance $R^2$ of 3% and 1%, respectively, but $RMSE_{0.66}$ improved by 19% and 33%, Fig. 3) or the rivers for Malmö ($R^2$ of 3%, $RMSE_{0.66}$ improved by 30%). These are not strongly correlated to other variables (Appendix Table A1 and A3, respectively), so this is not an artefact of our method. More surprisingly, there are also six cases where the north-westerly winds are the main contributors to the time series, as described in the previous paragraph, yet removing them improves the prediction of the highest sea level values. For $v_-$, those are Den Helder, Gedser, and Malmö; for $u_+$, Göteborg, Lowestoft, and Oslo (Fig. 3).

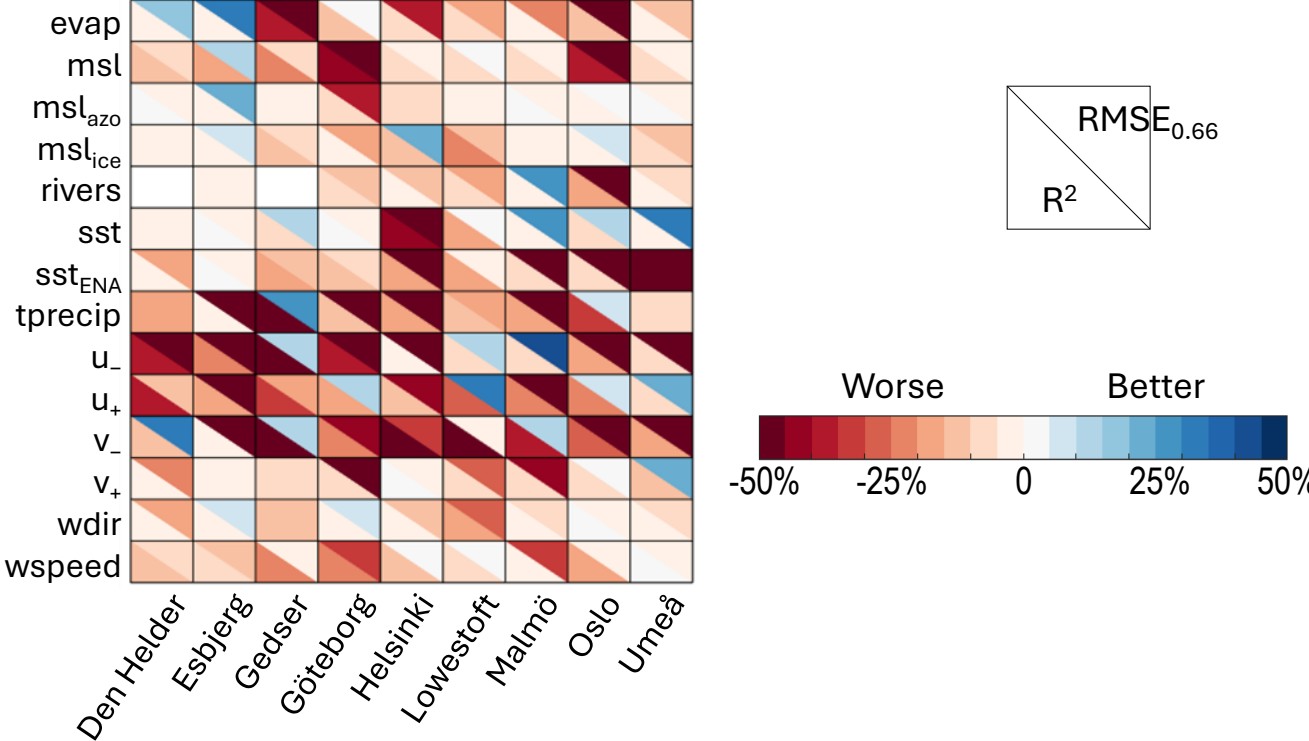

**Figure 3.** For each city on the x-axis, using the LSTM, change in explained variance ($R^2$, bottom left) and in root mean square error of the normalised sea level values larger than 0.66 ($RMSE_{0.66}$, top right) between the default run with all predictors and that where the one predictor on the y-axis was set to random values. Red means that the sea level prediction is worse when the predictor is randomised; blue that the prediction is improved. See Table 2 for the predictor definitions. For readability, we actually show - $RMSE_{0.66}$ normalised by the value of the default run. See Appendix Tables A4 - A6 for the actual values.

Our results suggest that periods of prolonged sea level and peaks in sea level have different drivers, at least in our region of interest. We investigate this further in the next section, focussing on the individual most extreme values in sea level for each
city. We also move to a method more adapted to point measurements: Random Forest regression.

### 3.2    The main drivers of the most extreme peaks depend on the coastline's geometry

For the extreme peaks of sea level, as expected from the literature (see Introduction), the performance of the Random Forest when including all predictors is slightly less than that of the LSTM for the prolonged periods of high sea level. The average root mean square error is less than 20 cm for all cities (Appendix Table A7) compared to an average non-normalised sea level
anomaly of more than 1 m. The correlation between test and predicted series is around 0.8, except for Helsinki and Oslo where

| City | $u_+$ median (max) | $v_-$ median (max) |
|---|---|---|
| Den Helder | $0.17 \pm 0.15$ (0.92) | $0.051 \pm 0.09$ (0.78) |
| Esbjerg | $0.14 \pm 0.14$ (0.88) | $0.001 \pm 0.07$ (0.63) |
| Gedser | $0.27 \pm 0.17$ (0.89) | $0.003 \pm 0.14$ (0.86) |
| Göteborg | $0.13 \pm 0.14$ (0.81) | $0.002 \pm 0.08$ (0.67) |
| Helsinki | $0.16 \pm 0.15$ (0.90) | $0.002 \pm 0.13$ (0.67) |
| Lowestoft | $0.22 \pm 0.16$ (0.91) | $0.003 \pm 0.11$ (0.72) |
| Malmö | $0.30 \pm 0.15$ (0.78) | $0.002 \pm 0.08$ (0.59) |
| Oslo | $0.05 \pm 0.10$ (0.72) | $0.005 \pm 0.07$ (0.91) |
| Umeå | $0.12 \pm 0.15$ (0.82) | $0.000 \pm 0.15$ (0.72) |

**Table 3.** For each city, median $\pm$ standard deviation, and maximum value in parentheses, of the normalised westerly $u_+$ and northerly $v_-$ wind subsets of the complete time series used as predictors for the LSTM. The normalisation was done on the complete time series.

it is at 0.7. This lower correlation is most likely because the sea level deep in these cities' respective bay and fjord is influenced by local weather processes that are not captured by ERA5's comparatively coarse resolution.

For the extreme peaks of sea level at all locations, there is no predictor in the Random Forest model that stands out as most important (Fig. 4), unlike the results for the periods of high sea level. In fact, the most important drivers at each locations seem related to the local coastline geometry and geography:

– For Den Helder, Esbjerg, Göteborg and Lowestoft, the westerly wind ($u_+$) is most important. This is probably because all these locations are relatively close to the source of the main Atlantic storms, especially so Lowestoft. For Den Helder and Esbjerg, this finding is in agreement with Sterlini et al. (2016) who argued that westerly winds indirectly matter because of induced Ekman transport that accumulates water on the coast. In the case of Esbjerg and Göteborg, the north-south orientation of the coastline also makes it vulnerable to westerly winds.

– For Gedser and Malmö, the northerly wind ($v_-$) is most important, which most likely is because it controls the flow of water between the Kattegat and the Baltic, or because both stations are protected from the westerlies by extensive land to the west (Fig. 1).

– For Helsinki and Oslo, it is the local sea level pressure (msl) that is most important, probably because both locations are located deep in their respective fjord / bay, and therefore either sheltered from the wind or the wind components from ERA5 are not representative of the extremely local weather that can develop in fjords.

– For Umeå, southerly wind ($v_+$) and precipitation are most important. Similar to Gedser and Malmö, the meridional wind most likely matters because Umeå lies on the west coast of its sea, and therefore is not affected by the westerly winds. Or, since the network also finds a high importance for precipitation, it could be because southerly winds bring warm moist air there.

Unlike for the LSTM, the values of the predictor variables at these locations are notably different during the peaks of extreme sea level compared to the rest of the time series (Table 4). The westerly wind is more than 5 times stronger than usual for Den Helder, Esbjerg, Göteborg and Lowestoft (medians larger than 6 m/s compared to the usual 1 m/s). For Gedser and Malmö, the northerly wind is more than twice as strong, reaching a median of 16.5 m/s for Gedser. The sea level pressure is anomalously low in Helsinki and Oslo, with medians lower than 1000 hPa. Umeå is the one city with anomalous southerly winds, more than 3 times as strong as usual. Additionally, for most cities, the wind speed is anomalously high (Table 4). This may be why the LSTM could not predict well the most extreme sea level values: the predictands have a different distribution during these peaks.

In agreement with Sterlini et al. (2017), the steric component, here represented by the local sea surface temperature (sst, Fig. 4), is important for some locations, but not as important as the meteorological component. Similarly, the remote drivers seem more important than for the periods of prolonged sea level: both $\mathrm{msl_{azo}}$ and $\mathrm{msl_{ice}}$ have some importance at all locations, which is consistent with the importance of their combined index, the North Atlantic Oscillation, for extremes in the region (Hurrell, 1995; Melet et al., 2024). $\mathrm{sst_{ENA}}$, which can be considered as a proxy for overall warming, also has a relative importance of more than 20% for more than half of the locations, which is consistent with the local relationship between global warming, mean sea level value, and extreme sea levels (Vousdoukas et al., 2017).

| City | msl | | $u_+$ | | $v_-$ | | $v_+$ | | wspeed | |
|---|---|---|---|---|---|---|---|---|---|---|
| | RF | All | RF | All | RF | All | RF | All | RF | All |
| Den Helder | $1005 \pm 11$ | $1020 \pm 11$ | $9.8 \pm 4.0$ | $\pm 1.3\ 3.9$ | $-2.0 \pm 4.8$ | $1.0 \pm 2.9$ | $2.5 \pm 3.4$ | $1.9 \pm 3.3$ | $11.6 \pm 3.1$ | $3.5 \pm 3.4$ |
| Esbjerg | $1001 \pm 11$ | $1021 \pm 11$ | $7.2 \pm 3.3$ | $\pm 0.9\ 2.9$ | $-1.6 \pm 2.6$ | $-1.6 \pm 1.9$ | $0.4 \pm 3.9$ | $1.3 \pm 2.4$ | $9.8 \pm 2.6$ | $3.4 \pm 2.5$ |
| Gedser | $1017 \pm 13$ | $1017 \pm 10$ | $3.2 \pm 2.4$ | $\pm 1.3\ 3.7$ | $-16.5 \pm 3.7$ | $-9.3 \pm 2.1$ | $2.1 \pm 0.8$ | $1.3 \pm 2.6$ | $14.3 \pm 2.9$ | $10.5 \pm 3.0$ |
| Göteborg | $999 \pm 11$ | $1021 \pm 12$ | $7.9 \pm 4.0$ | $\pm 1.3\ 2.9$ | $-2.4 \pm 2.2$ | $-2.4 \pm 1.7$ | $0.3 \pm 3.5$ | $0.9 \pm 2.5$ | $10.9 \pm 2.7$ | $4.4 \pm 2.5$ |
| Helsinki | $996 \pm 13$ | $1020 \pm 12$ | $3.9 \pm 2.7$ | $\pm 0.7\ 1.9$ | $-1.6 \pm 1.5$ | $-1.7 \pm 1.5$ | $1.1 \pm 3.4$ | $0.9 \pm 1.9$ | $7.3 \pm 2.3$ | $3.7 \pm 1.7$ |
| Lowestoft | $1016 \pm 11$ | $1020 \pm 11$ | $6.5 \pm 3.4$ | $\pm 0.6\ 3.1$ | $-4.6 \pm 4.8$ | $-0.6 \pm 2.8$ | $2.6 \pm 1.9$ | $1.5 \pm 3.2$ | $6.9 \pm 3.2$ | $2.6 \pm 2.9$ |
| Malmö | $1017 \pm 13$ | $1017 \pm 11$ | $1.6 \pm 2.6$ | $\pm 1.0\ 2.7$ | $-9.4 \pm 2.6$ | $-5.7 \pm 1.5$ | $1.6 \pm 1.9$ | $1.0 \pm 2.1$ | $7.7 \pm 2.5$ | $6.5 \pm 2.3$ |
| Oslo | $998 \pm 11$ | $1024 \pm 12$ | $1.1 \pm 1.8$ | $\pm 0.3\ 1.0$ | $-1.4 \pm 0.8$ | $-1.5 \pm 1.3$ | $2.2 \pm 2.5$ | $1.0 \pm 1.5$ | $4.5 \pm 2.0$ | $2.1 \pm 1.3$ |
| Umeå | $994 \pm 13$ | $1020 \pm 12$ | $4.3 \pm 4.2$ | $\pm 1.2\ 2.4$ | $0.0 \pm 2.2$ | $0.0 \pm 2.6$ | $6.9 \pm 4.3$ | $2.0 \pm 3.0$ | $9.2 \pm 3.0$ | $4.0 \pm 2.6$ |

**Table 4.** For each city, median $\pm$ standard deviation of the sea level pressure (msl, hPa), westerly ($u_+$, m/s), northerly ($v_-$, m/s), southerly ($v_+$, m/s) winds and wind speed (wspeed, m/s) for the subsets of the complete time series used as predictors for the Random Forest (columns "RF", left) compared to the complete time series (columns "All", right).

Finally, it is worth noting that the LSTM results when using the error on the high sea level values ($\mathrm{RMSE_{0.66}}$ on Fig. 3) and the Random Forest results (Fig. 4) agree quite well. There is a minority of cases where removing the predictor from the LSTM improved the performance, even though that predictor is deemed important by the Random Forest (blue triangles on Fig. 3 but red boxes on Fig. 4, e.g. the sea level pressure for Esbjerg or westerly wind for Lowestoft). But in most cases, either the two

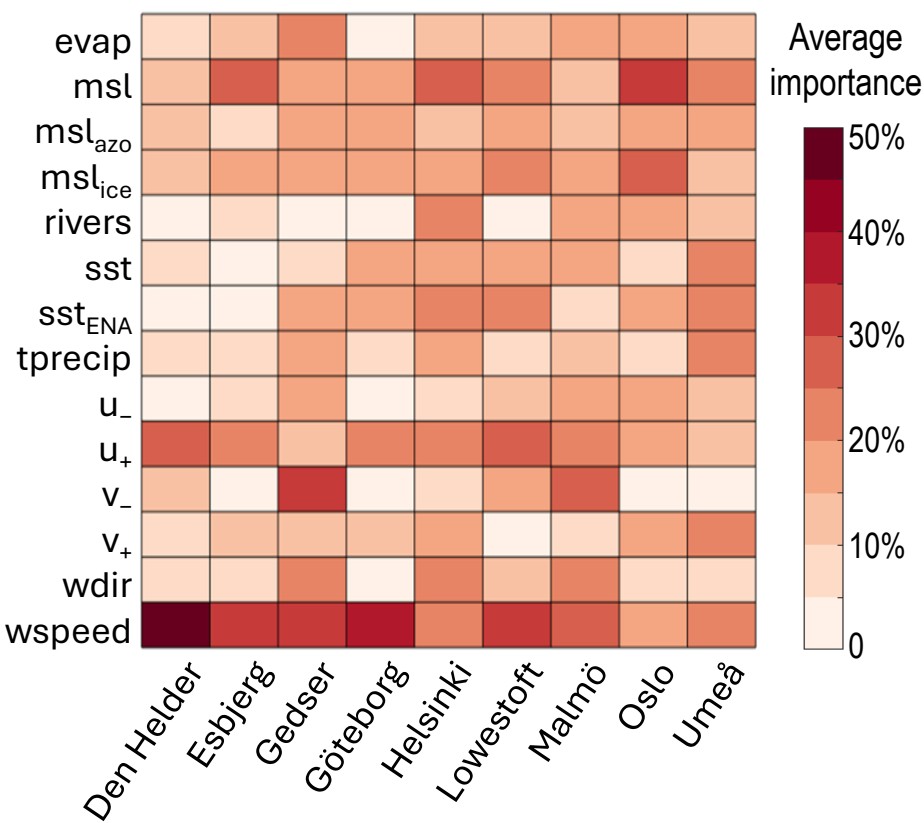

**Figure 4.** For each city on the x-axis, using the Random Forest, for each predictor on the y-axis, average importance over the 100 runs, summed for all the predictors' delays (i.e. simultaneous value and 9 delays). See Table 2 for the predictor definitions. See Appendix Tables A8 to A12 for the individual values for each delay rather than the sum, and the ensemble standard deviations.

methods agree that the predictor is important for extreme values (red on both figures, e.g. rivers for Helsinki or wind speed for Malmö), or they both agree that it does not and can/should be removed from the prediction (blue on Fig. 3 and pale on Fig. 4, e.g. evaporation in Den Helder or $\mathrm{msl_{azo}}$ in Esbjerg). Therefore, the two methods are more complementary than it first seemed, as long as one chooses the most relevant evaluation metric for the LSTM.

### 3.3 Applicability to sea level monitoring

We confirmed with a data-driven approach, and at a higher temporal resolution, that the hydrodynamics models were correct: extreme sea level around Northern Europe is primarily a result of the westerly winds (Melet et al., 2024, and references therein). This is excellent news since hydrodynamics models, despite their many property- and process-biases, are to-date the best tools for projecting future sea level (Vousdoukas et al., 2017) and its impacts (van de Wal et al., 2024), and therefore to inform policy makers. By developing two machine-learning based methods for different situations, we provide a cheaper and faster

(Hieronymus et al., 2019) alternative for shorter term decisions. Although we did not test this here, it should be feasible to detect an upcoming extreme peak during a period of high sea level using the disagreement in predictor importance between the two methods during such peaks. Since we worked with the non-tidal residuals of sea level, this method should remain functioning even as the background sea level increases (Melet et al., 2024) and as tides keep on changing non-linearly (e.g. Idier et al., 2017; Schindelegger et al., 2018), as long as no tipping point of the climate system changes the importance of the extreme sea level drivers. One might however have to change the definition of the remote drivers as air and oceans warm and storm tracks shift (Shaw et al., 2016). Future work could also consider developing a hybrid LSTM-Random Forest model for extreme sea level, as has been done recently for weather forecasting (Magesh et al., 2024).

Although this is common practice (e.g. Hieronymus et al., 2019; Ishida et al., 2020; Jiang et al., 2022b), a limitation of this study is the use of reanalysis data instead of meteorological observations. Reanalyses, and ERA5 in particular, are known to underestimate extreme values (Bell et al., 2021), and they provide multi-kilometer average values instead of those at the location of the tide gauge station. Unfortunately, most tide gauge stations do not have colocated meteorological observations. This is even worse for hydrological observations, which are not at the same location and, in this study, daily averages instead of instantaneous values. This is probably the reason why, surprisingly and contrary to common knowledge, our models found that rivers are not important for extreme sea level. There are also observations that we wish we could have included but, to the best of our knowledge, do not exist over the long time period needed for the training, such as the sea ice for Umeå. This could explain why our predictions never explained all the variance in the time series. However, since all our LSTM predictions explained more than two thirds of the signal, we are confident that we included the main predictors. Relocating or creating new observation stations is a policy decision. Policy makers should also consider whether to relocate or create new tide gauge stations, as the current ones often are at locations sheltered from the waves (Melet et al., 2024). We experienced that they sometimes are located so deep in the city centre that the sea level becomes too artificial to be predicted by atmospheric variables, such as Bergen on the Norwegian southwest coast (not shown). Depending on local vulnerabilities or flood defense systems, other locations on the nearby coast might be more representative (van de Wal et al., 2024).

Although our data covered three seas and some stations were relatively close to each other, we did not find any regional coherence at the hourly time scale, unlike what was found by Poropat et al. (2024) for the monthly variability. In fact, locations with similar coastline geometry behaved most similarly. We acknowledge that our region is comparatively large for a machine learning study where single tide gauge (Ishida et al., 2020) or single sea (Ayinde et al., 2023; Ruić et al., 2024) series are the norm, but still too small to dare extrapolating our findings too much. In the Baltic, Hieronymus et al. (2019) and Barzandeh et al. (2024) demonstrated the potential of LSTMs and CNNs on the west and east coast, respectively. But over a larger scale, data scarcity remains the limiting factor. Although the RMSE was significantly larger for Random Forest, amounting to up to 20% of the sea level value compared to less than 5% for LSTM, we still recommend using Random Forest when tide gauge observations are short and/or patchy, as we showed that Random Forest can work well with individual points, are extremely fast to train, and mostly agreed with the LSTM findings.

## 4 Conclusions

We used explainable AI to identify the drivers of extreme sea level events around Northern Europe from hourly tide gauge
data, hourly reanalysis meteorological time series, and daily river runoff. We found that periods of high sea level are driven by
westerly winds, but the short-lived peaks of highest sea level values depend on the local coastline geometry. To the best of our
knowledge, this is the first data-driven confirmation of the results found by hydrodynamic models (as reviewed in Melet et al.,
2024). This means that despite their many biases, these model-based projections of future sea level can be trusted for policy
making. Our results also potentially open the way for physics-informed machine-learning based sea level predictions.

We found that the more advanced Long Short Term Memory recurrent neural network performed best, with a correlation
with the test time series exceeding 0.8 and a low RMSE, yet the simpler Random Forest, despite its higher RMSE, performs
well enough to predict and explain the most extreme sea level values. That is, Random Forest is suitable for locations with short
and/or incomplete sea level time series. This is good news as there is no obvious reason why our models could not be used, with
minimum re-training and the potential addition of more relevant remote drivers (e.g. one indicative of cyclone formation at low
latitudes), in other parts of the world regardless of the status of their tide gauge network. As Europe's and the world's coastlines
vulnerability to extreme sea level will only increase with ongoing global warming-induced sea level rise (Vousdoukas et al.,
2017; van de Wal et al., 2024), if priority is not given to developing a better monitoring station network, implementing this
simple Random Forest method could be an easy and low-cost way to detect and prepare for upcoming peaks in sea level.

*Code and data availability.* The codes will be made available via Zenodo when the manuscript is closer to acceptance. The tide gauge data
for Sweden are freely available via the Swedish Meteorological and Hydrological Institute's website: https://www.smhi.se/data/oceanografi/
ladda-ner-oceanografiska-observationer/ (last accessed 5 Nov 2024). The tide gauge data for the other cities come from the Global Extreme
Sea Level Analysis (GESLA) dataset version 3 (Woodworth et al., 2016; Haigh et al., 2023), freely available via https://gesla787883612.
wordpress.com/ (last accessed 5 Nov 2024). The reanalysis data from ERA5 (Hersbach et al., 2020) are freely available via the Copernicus
Climate Data Store: https://cds.climate.copernicus.eu/datasets/reanalysis-era5-single-levels (last accessed 2 Jan 2025). The river runoff data
are provided by the Global Runoff Data Centre (GRDC) and are freely available via https://grdc.bafg.de/ (last accessed 27 Nov 2024).

| | msl | msl$_{azo}$ | msl$_{ice}$ | rivers | sst | sst$_{ENA}$ | tprecip | u$_-$ | u$_+$ | v$_-$ | v$_+$ | wdir | wspeed |
|---|---|---|---|---|---|---|---|---|---|---|---|---|---|
| | | | | | | Den Helder | | | | | | | |
| evap | -26 | 14 | 16 | - | 34 | 26 | - | 17 | 33 | 19 | -8 | 3 | 28 |
| msl | | -4 | 8 | - | 7 | 15 | -16 | 2 | -32 | - | -26 | -8 | -49 |
| msl$_{azo}$ | | | -8 | - | -22 | -29 | 3 | -14 | 20 | 11 | -4 | 17 | 19 |
| msl$_{ice}$ | | | | - | 22 | 32 | -5 | 17 | 4 | 19 | -36 | - | -21 |
| rivers | | | | - | - | - | - | - | - | - | - | - | - |
| sst | | | | | 80 | - | 18 | - | - | -8 | -12 | -8 |
| sst$_{ENA}$ | | | | | | - | 26 | -7 | 2 | -13 | -15 | -15 |
| tprecip | | | | | | | - | - | -6 | 16 | -6 | 10 |
| u$_-$ | | | | | | | | -13 | - | -8 | -64 | -18 |
| u$_+$ | | | | | | | | | - | -24 | 31 | 67 |
| v$_-$ | | | | | | | | | | -14 | 23 | 12 |
| v$_+$ | | | | | | | | | | | -27 | 43 |
| wdir | | | | | | | | | | | | 21 |
| | | | | | | Esbjerg | | | | | | | |
| evap | -8 | -6 | 4 | -15 | 25 | 25 | 6 | -18 | - | -10 | 12 | 5 | 6 |
| msl | | -4 | 23 | -45 | 5 | 4 | -13 | 12 | -36 | - | -29 | -10 | -56 |
| msl$_{azo}$ | | | -23 | 12 | - | -32 | 4 | -18 | 12 | -4 | 5 | 15 | 14 |
| msl$_{ice}$ | | | | -21 | 17 | 25 | -16 | - | - | 13 | -33 | 10 | -26 |
| rivers | | | | | -38 | -48 | 8 | -22 | 27 | 11 | 10 | 28 | 35 |
| sst | | | | | | 82 | - | -22 | - | - | - | 14 | - |
| sst$_{ENA}$ | | | | | | | - | -8 | -11 | -6 | - | - | -13 |
| tprecip | | | | | | | | -6 | -9 | -11 | 22 | -6 | 10 |
| u$_-$ | | | | | | | | | -11 | -10 | -12 | -71 | -11 |
| u$_+$ | | | | | | | | | | 21 | -19 | 32 | 69 |
| v$_-$ | | | | | | | | | | | -17 | 42 | 11 |
| v$_+$ | | | | | | | | | | | | -23 | 47 |
| wdir | | | | | | | | | | | | | 19 |
| | | | | | | Gedser | | | | | | | |
| evap | -35 | - | 38 | - | 22 | 16 | -13 | 3 | 23 | 39 | - | 15 | 29 |
| msl | | -5 | -19 | - | - | -15 | -6 | -9 | -20 | - | -24 | 6 | -32 |
| msl$_{azo}$ | | | -4 | - | -20 | -36 | - | -7 | -5 | 3 | 9 | 4 | 4 |
| msl$_{ice}$ | | | | - | 10 | 24 | -8 | - | 15 | 24 | -22 | 11 | 9 |
| rivers | | | | | - | - | - | - | - | - | - | - | - |
| sst | | | | | | 54 | 13 | 3 | 5 | - | 3 | -8 | 8 |
| sst$_{ENA}$ | | | | | | | 12 | - | -9 | - | 6 | -11 | -6 |
| tprecip | | | | | | | | - | -13 | -12 | 9 | -9 | -8 |
| u$_-$ | | | | | | | | | -14 | - | 10 | -58 | -18 |
| u$_+$ | | | | | | | | | | -12 | -25 | 22 | 82 |
| v$_-$ | | | | | | | | | | | -13 | 38 | 5 |
| v$_+$ | | | | | | | | | | | | -35 | 16 |
| wdir | | | | | | | | | | | | | 23 |

**Table A1.** For each city, correlation (R, in %) between the time series of the predictors used for the LSTM. Only correlations significant at 95% are shown.

| | msl | msl$_{azo}$ | msl$_{ice}$ | rivers | sst | sst$_{ENA}$ | tprecip | u$_-$ | u$_+$ | v$_-$ | v$_+$ | wdir | wspeed |
|---|---|---|---|---|---|---|---|---|---|---|---|---|---|
| | | | | | | Göteborg | | | | | | | |
| evap | -31 | 19 | - | 19 | 5 | -7 | 7 | 6 | 14 | 10 | 12 | 3 | 26 |
| msl | | - | - | -56 | -5 | 16 | -12 | -24 | -17 | -8 | -8 | 6 | -20 |
| msl$_{azo}$ | | | -12 | -5 | 17 | - | - | - | 16 | -12 | 3 | - | 22 |
| msl$_{ice}$ | | | | -27 | -8 | 21 | -16 | - | 18 | 15 | -36 | 12 | -12 |
| rivers | | | | | 15 | -49 | 17 | 12 | 4 | - | 22 | -9 | 19 |
| sst | | | | | | -19 | - | - | 18 | -6 | - | 9 | 17 |
| sst$_{ENA}$ | | | | | | | - | -8 | -6 | 4 | -13 | 6 | -12 |
| tprecip | | | | | | | | 5 | -14 | -9 | 27 | -16 | 6 |
| u$_-$ | | | | | | | | | -12 | - | 8 | -53 | - |
| u$_+$ | | | | | | | | | | 4 | -39 | 43 | 65 |
| v$_-$ | | | | | | | | | | | -15 | 19 | -6 |
| v$_+$ | | | | | | | | | | | | -41 | 35 |
| wdir | | | | | | | | | | | | | 18 |
| | | | | | | Helsinki | | | | | | | |
| evap | 19 | -18 | -15 | 4 | - | - | 7 | -15 | - | -16 | -6 | 5 | -7 |
| msl | | -26 | -21 | -39 | -13 | -7 | -12 | -17 | -5 | - | -19 | 12 | -18 |
| msl$_{azo}$ | | | 8 | -7 | 23 | 16 | 8 | -11 | 6 | - | 10 | - | 13 |
| msl$_{ice}$ | | | | -11 | -15 | 9 | -10 | -10 | - | 20 | -11 | 11 | -5 |
| rivers | | | | | 15 | -31 | 9 | - | 14 | -8 | -8 | -7 | - |
| sst | | | | | | -39 | 8 | -19 | 20 | -13 | 14 | - | 16 |
| sst$_{ENA}$ | | | | | | | - | -7 | -15 | - | 5 | - | - |
| tprecip | | | | | | | | 7 | -7 | -14 | 24 | -12 | 13 |
| u$_-$ | | | | | | | | | -15 | -9 | - | -65 | -11 |
| u$_+$ | | | | | | | | | | -18 | -6 | 11 | 58 |
| v$_-$ | | | | | | | | | | | -14 | 39 | 5 |
| v$_+$ | | | | | | | | | | | | -21 | 62 |
| wdir | | | | | | | | | | | | | 9 |
| | | | | | | Lowestoft | | | | | | | |
| evap | -7 | -12 | 12 | -21 | 38 | 35 | - | - | 27 | 32 | - | 23 | 23 |
| msl | | - | 50 | -49 | 40 | 33 | -16 | -13 | -29 | 20 | -34 | 24 | -57 |
| msl$_{azo}$ | | | -45 | 32 | -35 | -48 | - | - | 13 | -15 | 6 | - | 17 |
| msl$_{ice}$ | | | | -47 | 48 | 47 | -12 | - | -16 | 33 | -24 | 20 | -34 |
| rivers | | | | | -62 | -66 | 10 | - | 27 | -17 | 25 | -9 | 42 |
| sst | | | | | | 94 | -7 | - | - | 19 | -25 | 13 | -28 |
| sst$_{ENA}$ | | | | | | | -7 | - | -8 | 17 | -31 | 13 | -31 |
| tprecip | | | | | | | | - | - | -10 | 20 | -19 | 12 |
| u$_-$ | | | | | | | | | - | - | -7 | -15 | - |
| u$_+$ | | | | | | | | | | - | -7 | 43 | 65 |
| v$_-$ | | | | | | | | | | | -12 | 53 | - |
| v$_+$ | | | | | | | | | | | | -35 | 61 |
| wdir | | | | | | | | | | | | | - |

**Table A2.** Appendix Table A1 continues

| | msl | msl$_{azo}$ | msl$_{ice}$ | rivers | sst | sst$_{ENA}$ | tprecip | u$_-$ | u$_+$ | v$_-$ | v$_+$ | wdir | wspeed |
|---|---|---|---|---|---|---|---|---|---|---|---|---|---|
| | | | | | | | Malmö | | | | | | |
| evap | 13 | -9 | - | -10 | -13 | -8 | - | - | -21 | 3 | -8 | 7 | -25 |
| msl | | -21 | -11 | -29 | 11 | -19 | -11 | -12 | -34 | - | -20 | 5 | -42 |
| msl$_{azo}$ | | | 11 | 3 | - | -5 | 3 | 5 | 4 | - | 13 | -4 | 12 |
| msl$_{ice}$ | | | | 21 | - | 15 | -11 | - | 15 | 30 | -29 | 28 | 4 |
| rivers | | | | | 22 | 52 | 16 | - | 11 | - | 7 | - | 11 |
| sst | | | | | | 23 | - | -9 | 11 | -5 | - | - | 8 |
| sst$_{ENA}$ | | | | | | | 9 | - | 4 | 4 | 3 | - | 4 |
| tprecip | | | | | | | | 7 | -10 | -11 | 20 | -15 | - |
| u$_-$ | | | | | | | | | -11 | 3 | - | -52 | -6 |
| u$_+$ | | | | | | | | | | - | -15 | 20 | 82 |
| v$_-$ | | | | | | | | | | | -14 | 43 | 7 |
| v$_+$ | | | | | | | | | | | | -33 | 33 |
| wdir | | | | | | | | | | | | | 10 |
| | | | | | | | Oslo | | | | | | |
| evap | 11 | - | -22 | 16 | - | - | 24 | - | -29 | -21 | 17 | -13 | 5 |
| msl | | -19 | 35 | -31 | -5 | 15 | -13 | -27 | -19 | - | -9 | 18 | -29 |
| msl$_{azo}$ | | | -17 | -18 | -9 | -20 | - | 4 | 10 | 4 | -4 | - | - |
| msl$_{ice}$ | | | | -26 | 5 | 27 | -18 | -16 | 9 | 26 | -25 | 27 | -28 |
| rivers | | | | | 17 | - | 22 | 28 | -16 | -16 | 19 | -21 | 23 |
| sst | | | | | | 88 | 5 | 8 | -8 | - | -5 | -6 | - |
| sst$_{ENA}$ | | | | | | | - | - | -9 | - | -11 | - | -13 |
| tprecip | | | | | | | | 26 | -14 | -8 | 7 | -21 | 11 |
| u$_-$ | | | | | | | | | -10 | - | -4 | -22 | 24 |
| u$_+$ | | | | | | | | | | -7 | -5 | 23 | 30 |
| v$_-$ | | | | | | | | | | | -30 | 18 | -21 |
| v$_+$ | | | | | | | | | | | | -14 | 82 |
| wdir | | | | | | | | | | | | | -15 |
| | | | | | | | Umeå | | | | | | |
| evap | -19 | 19 | 19 | 9 | 16 | 16 | -5 | - | 21 | 46 | -20 | 11 | 29 |
| msl | | -22 | -16 | - | -9 | - | -5 | -8 | -23 | -5 | - | - | -26 |
| msl$_{azo}$ | | | 19 | -11 | 20 | 18 | 8 | -8 | 14 | 8 | -6 | 9 | 10 |
| msl$_{ice}$ | | | | - | - | 11 | -7 | -19 | 11 | 22 | -26 | 11 | -6 |
| rivers | | | | | 39 | 12 | - | 7 | -6 | 5 | -4 | - | -6 |
| sst | | | | | | 64 | 8 | -9 | - | - | - | 8 | - |
| sst$_{ENA}$ | | | | | | | 5 | -8 | -5 | 7 | -10 | 7 | -9 |
| tprecip | | | | | | | | 16 | -12 | -14 | 27 | -13 | 12 |
| u$_-$ | | | | | | | | | -22 | -13 | 16 | -55 | - |
| u$_+$ | | | | | | | | | | - | -22 | 30 | 51 |
| v$_-$ | | | | | | | | | | | -51 | 28 | 10 |
| v$_+$ | | | | | | | | | | | | -34 | 49 |
| wdir | | | | | | | | | | | | | 11 |

**Table A3.** Appendix Table A1 continues.

| Predictor | Den Helder | Esbjerg | Gedser | Göteborg | Helsinki | Lowestoft | Malmö | Oslo | Umeå |
|---|---|---|---|---|---|---|---|---|---|
| default | 68.7 | 75.3 | 69.1 | 76.3 | 62.5 | 65.4 | 62.6 | 65.8 | 67.3 |
| evap | 65.6 | 74.5 | 30.9 | 63.9 | 52.5 | 57.8 | 62.3 | 54.7 | 66.1 |
| msl | 57.6 | 60.0 | 45.9 | 35.5 | 57.3 | 58.0 | 57.2 | 26.3 | 61.5 |
| $msl_{azo}$ | 69.8 | 72.7 | 68.2 | 71.1 | 57.3 | 60.8 | 63.4 | 65.0 | 67.3 |
| $msl_{ice}$ | 66.7 | 75.3 | 58.9 | 73.4 | 52.3 | 43.5 | 62.4 | 64.1 | 61.9 |
| rivers | 68.6 | 73.7 | 69.1 | 70.8 | 60.5 | 58.3 | 59.4 | 46.5 | 63.6 |
| sst | 68.6 | 75.9 | 63.2 | 76.3 | 18.0 | 45.9 | 59.7 | 60.7 | 62.8 |
| $sst_{ENA}$ | 68.7 | 76.0 | 49.4 | 65.4 | 44.7 | 47.8 | 54.6 | 57.1 | 15.8 |
| tprecip | 53.4 | 71.7 | 3.9 | 64.5 | 45.7 | 53.9 | 45.1 | 32.1 | 62.1 |
| $u_-$ | 30.5 | 51.7 | 20.0 | 37.2 | 58.3 | 59.3 | 52.7 | 48.8 | 57.7 |
| $u_+$ | 31.7 | 58.0 | 35.2 | 59.1 | 50.9 | 39.5 | 38.0 | 42.5 | 61.2 |
| $v_-$ | 55.0 | 71.1 | 13.9 | 52.5 | 16.3 | 9.9 | 26.1 | 36.6 | 48.5 |
| $v_+$ | 65.7 | 72.9 | 63.5 | 66.5 | 66.0 | 59.0 | 54.2 | 59.1 | 56.7 |
| wdir | 65.0 | 74.0 | 56.1 | 75.1 | 61.4 | 46.9 | 60.6 | 65.9 | 65.9 |
| wspeed | 57.9 | 69.5 | 46.1 | 53.0 | 50.5 | 56.7 | 59.4 | 48.5 | 67.5 |

**Table A4.** For each city, for the LSTM, explained variance ($R^2$, in %) of the default run with all predictors (top row) and of each run with feature permutation, where the predictor was set to random values.

| Predictor | Den Helder | Esbjerg | Gedser | Göteborg | Helsinki | Lowestoft | Malmö | Oslo | Umeå |
|---|---|---|---|---|---|---|---|---|---|
| default | 0.044 | 0.098 | 0.062 | 0.051 | 0.049 | 0.052 | 0.053 | 0.050 | 0.052 |
| evap | 0.048 | 0.080 | 0.187 | 0.120 | 0.070 | 0.050 | 0.054 | 0.094 | 0.059 |
| msl | 0.049 | 0.102 | 0.084 | 0.086 | 0.051 | 0.052 | 0.058 | 0.080 | 0.054 |
| $msl_{azo}$ | 0.043 | 0.087 | 0.063 | 0.057 | 0.052 | 0.052 | 0.050 | 0.050 | 0.056 |
| $msl_{ice}$ | 0.045 | 0.096 | 0.080 | 0.054 | 0.049 | 0.055 | 0.051 | 0.052 | 0.055 |
| rivers | 0.044 | 0.105 | 0.063 | 0.057 | 0.050 | 0.053 | 0.061 | 0.074 | 0.055 |
| sst | 0.044 | 0.107 | 0.076 | 0.051 | 0.158 | 0.057 | 0.063 | 0.062 | 0.052 |
| $sst_{ENA}$ | 0.048 | 0.112 | 0.141 | 0.062 | 0.120 | 0.054 | 0.186 | 0.074 | 0.119 |
| tprecip | 0.063 | 0.262 | 0.152 | 0.215 | 0.177 | 0.054 | 0.177 | 0.105 | 0.065 |
| $u_-$ | 0.094 | 0.105 | 0.141 | 0.190 | 0.096 | 0.055 | 0.098 | 0.091 | 0.083 |
| $u_+$ | 0.071 | 0.175 | 0.103 | 0.084 | 0.058 | 0.067 | 0.095 | 0.124 | 0.042 |
| $v_-$ | 0.054 | 0.140 | 0.140 | 0.077 | 0.090 | 0.084 | 0.191 | 0.108 | 0.093 |
| $v_+$ | 0.051 | 0.092 | 0.078 | 0.063 | 0.053 | 0.056 | 0.080 | 0.055 | 0.046 |
| wdir | 0.048 | 0.096 | 0.099 | 0.053 | 0.058 | 0.058 | 0.058 | 0.051 | 0.054 |
| wspeed | 0.050 | 0.141 | 0.087 | 0.074 | 0.043 | 0.053 | 0.055 | 0.065 | 0.046 |

**Table A5.** Same as Appendix Table A4 but for the overall root mean square error.

| Predictor | Den Helder | Esbjerg | Gedser | Göteborg | Helsinki | Lowestoft | Malmö | Oslo | Umeå |
|---|---|---|---|---|---|---|---|---|---|
| default | 0.024 | 0.042 | 0.031 | 0.026 | 0.021 | 0.016 | 0.012 | 0.019 | 0.032 |
| evap | 0.019 | 0.029 | 0.049 | 0.025 | 0.029 | 0.019 | 0.015 | 0.039 | 0.036 |
| msl | 0.025 | 0.031 | 0.033 | 0.041 | 0.022 | 0.016 | 0.013 | 0.028 | 0.033 |
| $\text{msl}_{\text{azo}}$ | 0.020 | 0.034 | 0.032 | 0.036 | 0.023 | 0.016 | 0.009 | 0.018 | 0.035 |
| $\text{msl}_{\text{ice}}$ | 0.024 | 0.041 | 0.033 | 0.030 | 0.016 | 0.018 | 0.013 | 0.018 | 0.036 |
| rivers | 0.025 | 0.043 | 0.031 | 0.029 | 0.024 | 0.019 | 0.009 | 0.035 | 0.034 |
| sst | 0.024 | 0.047 | 0.026 | 0.029 | 0.050 | 0.016 | 0.009 | 0.016 | 0.021 |
| $\text{sst}_{\text{ENA}}$ | 0.028 | 0.047 | 0.034 | 0.028 | 0.058 | 0.017 | 0.038 | 0.031 | 0.081 |
| tprecip | 0.028 | 0.114 | 0.022 | 0.044 | 0.051 | 0.019 | 0.019 | 0.018 | 0.035 |
| $\text{u}_{-}$ | 0.058 | 0.067 | 0.026 | 0.076 | 0.038 | 0.014 | 0.007 | 0.038 | 0.050 |
| $\text{u}_{+}$ | 0.025 | 0.063 | 0.035 | 0.023 | 0.030 | 0.010 | 0.026 | 0.017 | 0.025 |
| $\text{v}_{-}$ | 0.016 | 0.080 | 0.027 | 0.038 | 0.027 | 0.016 | 0.011 | 0.042 | 0.062 |
| $\text{v}_{+}$ | 0.029 | 0.041 | 0.032 | 0.041 | 0.023 | 0.020 | 0.018 | 0.019 | 0.025 |
| wdir | 0.027 | 0.041 | 0.034 | 0.024 | 0.023 | 0.021 | 0.013 | 0.020 | 0.034 |
| wspeed | 0.025 | 0.048 | 0.032 | 0.034 | 0.020 | 0.016 | 0.017 | 0.019 | 0.029 |

**Table A6.** Same as Appendix Table A4 but for the root mean square error of the normalised sea level values larger than 0.66 ($\text{RMSE}_{0.66}$).

| | Den Helder | Esbjerg | Gedser | Göteborg | Helsinki | Lowestoft | Malmö | Oslo | Umeå |
|---|---|---|---|---|---|---|---|---|---|
| RMSE | 0.14 | 0.19 | 0.13 | 0.10 | 0.10 | 0.19 | 0.13 | 0.12 | 0.08 |
| Corr (R) | 0.86 | 0.77 | 0.79 | 0.79 | 0.70 | 0.81 | 0.76 | 0.72 | 0.78 |

**Table A7.** For each city, for the Random Forest, mean performance over 100 runs with all predictors: root mean square error (RMSE, in m) and correlation between the test set and its predicted values.

| Predictor | Delay (h) | Den Helder | Esbjerg | Gedser | Göteborg | Helsinki | Lowestoft | Malmö | Oslo | Umeå |
|---|---|---|---|---|---|---|---|---|---|---|
| evap | 0 | - | $1.4 \pm 0.1$ | $2.0 \pm 0.2$ | - | - | $2.0 \pm 0.3$ | $2.0 \pm 0.2$ | $1.9 \pm 0.3$ | - |
| | 1 | - | $1.5 \pm 0.2$ | $2.0 \pm 0.3$ | - | - | $1.9 \pm 0.3$ | $1.9 \pm 0.3$ | $2.1 \pm 0.3$ | - |
| | 3 | $1.3 \pm 0.1$ | $1.7 \pm 0.2$ | $2.1 \pm 0.4$ | - | - | $1.9 \pm 0.2$ | $2.1 \pm 0.3$ | $1.9 \pm 0.3$ | - |
| | 6 | $1.3 \pm 0.2$ | $1.8 \pm 0.2$ | $2.7 \pm 0.5$ | $1.7 \pm 0.3$ | - | $2.1 \pm 0.3$ | $2.1 \pm 0.3$ | - | $1.5 \pm 0.0$ |
| | 12 | $1.3 \pm 0.2$ | $1.6 \pm 0.2$ | $3.4 \pm 0.6$ | - | $2.1 \pm 0.4$ | $2.6 \pm 0.5$ | $2.0 \pm 0.4$ | $2.0 \pm 0.0$ | $2.0 \pm 0.5$ |
| | 24 | - | $1.4 \pm 0.1$ | $1.8 \pm 0.4$ | - | $1.9 \pm 0.4$ | - | $2.1 \pm 0.3$ | - | $2.2 \pm 0.5$ |
| | 48 | - | - | $2.1 \pm 0.2$ | - | $2.5 \pm 0.7$ | $2.0 \pm 0.2$ | $2.1 \pm 0.3$ | $1.8 \pm 0.1$ | $1.9 \pm 0.0$ |
| | 72 | - | - | $2.2 \pm 0.3$ | - | $3.6 \pm 1.2$ | $2.0 \pm 0.3$ | $1.4 \pm 0.2$ | $1.7 \pm 0.2$ | - |
| | 120 | - | $1.5 \pm 0.1$ | $2.1 \pm 0.3$ | - | $2.2 \pm 0.5$ | - | $1.9 \pm 0.1$ | $1.7 \pm 0.2$ | $1.7 \pm 0.1$ |
| | 168 | $1.3 \pm 0.3$ | $1.1 \pm 0.0$ | $2.1 \pm 0.3$ | - | $2.2 \pm 0.5$ | - | $2.2 \pm 0.4$ | $4.2 \pm 1.1$ | $1.8 \pm 0.3$ |
| msl | 0 | $1.3 \pm 0.1$ | $3.2 \pm 0.4$ | $2.0 \pm 0.3$ | $2.7 \pm 0.5$ | $2.9 \pm 0.8$ | $2.3 \pm 0.2$ | $2.0 \pm 0.3$ | $6.1 \pm 0.9$ | $2.6 \pm 0.6$ |
| | 1 | $1.3 \pm 0.1$ | $3.6 \pm 0.5$ | $2.2 \pm 0.0$ | $2.9 \pm 0.6$ | $3.1 \pm 0.8$ | $2.4 \pm 0.3$ | $1.9 \pm 0.2$ | $5.2 \pm 0.9$ | $2.8 \pm 0.8$ |
| | 3 | $1.4 \pm 0.2$ | $4.4 \pm 0.6$ | $1.9 \pm 0.2$ | $2.3 \pm 0.5$ | $3.8 \pm 0.9$ | $2.2 \pm 0.2$ | - | $3.4 \pm 0.6$ | $2.2 \pm 0.4$ |
| | 6 | $1.6 \pm 0.2$ | $4.9 \pm 0.6$ | - | $2.6 \pm 0.7$ | $4.4 \pm 1.0$ | $2.2 \pm 0.2$ | $1.6 \pm 0.3$ | $2.6 \pm 0.5$ | $2.2 \pm 0.5$ |
| | 12 | $1.8 \pm 0.2$ | $3.9 \pm 0.6$ | $2.1 \pm 0.3$ | $3.5 \pm 0.9$ | $2.1 \pm 0.4$ | $2.0 \pm 0.2$ | $2.2 \pm 0.3$ | $3.2 \pm 0.7$ | $3.0 \pm 0.8$ |
| | 24 | - | $2.0 \pm 0.3$ | $2.2 \pm 0.3$ | $2.4 \pm 0.4$ | $2.2 \pm 0.5$ | $2.7 \pm 0.4$ | $2.1 \pm 0.3$ | $1.9 \pm 0.3$ | $2.0 \pm 0.4$ |
| | 48 | - | $1.7 \pm 0.2$ | $1.7 \pm 0.0$ | $1.4 \pm 0.2$ | $3.3 \pm 1.2$ | $2.5 \pm 0.4$ | $2.0 \pm 0.3$ | $2.1 \pm 0.4$ | $1.9 \pm 0.5$ |
| | 72 | $1.3 \pm 0.1$ | $1.6 \pm 0.2$ | - | - | $4.4 \pm 1.3$ | $2.2 \pm 0.3$ | - | $1.8 \pm 0.2$ | $2.3 \pm 0.5$ |
| | 120 | - | - | $1.9 \pm 0.3$ | $1.4 \pm 0.3$ | $2.2 \pm 0.5$ | $2.2 \pm 0.3$ | - | $1.9 \pm 0.3$ | $2.1 \pm 0.5$ |
| | 168 | $1.4 \pm 0.2$ | - | $1.9 \pm 0.3$ | - | $1.6 \pm 0.2$ | $3.0 \pm 0.5$ | $2.6 \pm 0.5$ | $1.9 \pm 0.2$ | $2.1 \pm 0.5$ |
| $msl_{azo}$ | 0 | $1.3 \pm 0.1$ | - | $2.0 \pm 0.0$ | $1.5 \pm 0.2$ | - | $2.0 \pm 0.2$ | $1.8 \pm 0.0$ | - | $1.8 \pm 0.5$ |
| | 1 | $1.3 \pm 0.1$ | - | $1.8 \pm 0.0$ | $1.1 \pm 0.0$ | $1.5 \pm 0.0$ | $2.1 \pm 0.2$ | - | - | $1.8 \pm 0.3$ |
| | 3 | $1.3 \pm 0.1$ | - | $1.8 \pm 0.2$ | $1.8 \pm 0.4$ | $1.9 \pm 0.0$ | $2.0 \pm 0.2$ | - | - | $1.8 \pm 0.3$ |
| | 6 | $1.3 \pm 0.2$ | - | $1.9 \pm 0.3$ | $1.6 \pm 0.0$ | - | $2.1 \pm 0.2$ | - | $1.8 \pm 0.3$ | - |
| | 12 | $1.3 \pm 0.2$ | - | $1.9 \pm 0.0$ | $2.0 \pm 0.4$ | $1.8 \pm 0.6$ | $2.2 \pm 0.2$ | $2.3 \pm 0.3$ | $2.1 \pm 0.5$ | $1.2 \pm 0.0$ |
| | 24 | $1.3 \pm 0.2$ | $1.6 \pm 0.2$ | $2.1 \pm 0.3$ | $1.9 \pm 0.4$ | $2.0 \pm 0.3$ | $2.1 \pm 0.3$ | $2.1 \pm 0.3$ | $2.7 \pm 0.6$ | $1.8 \pm 0.3$ |
| | 48 | $1.7 \pm 0.3$ | $1.6 \pm 0.2$ | - | $2.3 \pm 0.5$ | $1.5 \pm 0.0$ | $2.2 \pm 0.3$ | $2.4 \pm 0.3$ | $3.0 \pm 0.7$ | $3.4 \pm 0.9$ |
| | 72 | $1.4 \pm 0.2$ | $1.7 \pm 0.2$ | - | $1.7 \pm 0.2$ | - | $2.0 \pm 0.2$ | $2.2 \pm 0.3$ | $1.9 \pm 0.2$ | $3.0 \pm 1.0$ |
| | 120 | - | $1.5 \pm 0.2$ | $2.1 \pm 0.3$ | $1.7 \pm 0.3$ | $2.0 \pm 0.5$ | - | $2.1 \pm 0.3$ | $2.0 \pm 0.2$ | $3.0 \pm 0.9$ |
| | 168 | - | $1.5 \pm 0.0$ | $2.2 \pm 0.3$ | $1.7 \pm 0.4$ | $2.2 \pm 0.5$ | $1.9 \pm 0.0$ | $1.9 \pm 0.2$ | $2.0 \pm 0.3$ | $1.9 \pm 0.3$ |

**Table A8.** For each city, for the Random Forest, average feature importance and its standard deviation (in %) across the 100 ensemble members of the Random Forest regression, for each tested delay (second column) of each predictor (first column). No value means that the feature was never selected as importance, see Methods.

| Predictor | Delay (h) | Den Helder | Esbjerg | Gedser | Göteborg | Helsinki | Lowestoft | Malmö | Oslo | Umeå |
|---|---|---|---|---|---|---|---|---|---|---|
| msl$_{ice}$ | 0 | $1.2 \pm 0.1$ | $1.5 \pm 0.0$ | - | $1.7 \pm 0.3$ | $2.2 \pm 0.7$ | $2.2 \pm 0.3$ | - | $2.3 \pm 0.4$ | - |
| | 1 | $1.3 \pm 0.2$ | $1.5 \pm 0.1$ | $1.6 \pm 0.0$ | $1.6 \pm 0.3$ | $2.4 \pm 0.5$ | $2.4 \pm 0.3$ | $2.0 \pm 0.2$ | $2.2 \pm 0.3$ | - |
| | 3 | $1.3 \pm 0.1$ | $1.4 \pm 0.2$ | $1.9 \pm 0.2$ | $1.9 \pm 0.4$ | $1.6 \pm 0.2$ | $2.7 \pm 0.4$ | $2.0 \pm 0.2$ | $2.2 \pm 0.4$ | - |
| | 6 | $1.2 \pm 0.1$ | $1.5 \pm 0.2$ | $2.0 \pm 0.3$ | $2.3 \pm 0.4$ | $1.9 \pm 0.2$ | $2.7 \pm 0.4$ | $1.8 \pm 0.0$ | $2.7 \pm 0.5$ | $2.5 \pm 0.0$ |
| | 12 | $1.3 \pm 0.1$ | $2.0 \pm 0.3$ | $2.2 \pm 0.3$ | $2.6 \pm 0.6$ | $2.0 \pm 0.3$ | $2.0 \pm 0.4$ | $2.1 \pm 0.3$ | $2.0 \pm 0.3$ | - |
| | 24 | $1.4 \pm 0.2$ | $3.1 \pm 0.5$ | $2.1 \pm 0.3$ | $2.1 \pm 0.4$ | $2.3 \pm 0.5$ | $2.3 \pm 0.3$ | $1.9 \pm 0.2$ | $2.0 \pm 0.3$ | - |
| | 48 | - | $1.9 \pm 0.3$ | $2.1 \pm 0.3$ | $1.6 \pm 0.3$ | - | $1.9 \pm 0.2$ | $1.9 \pm 0.2$ | $1.9 \pm 0.2$ | $2.1 \pm 0.0$ |
| | 72 | $1.1 \pm 0.0$ | $1.8 \pm 0.2$ | $2.1 \pm 0.3$ | $2.2 \pm 0.4$ | $2.1 \pm 0.5$ | - | $2.3 \pm 0.4$ | $6.5 \pm 1.5$ | $1.9 \pm 0.3$ |
| | 120 | $1.3 \pm 0.2$ | $1.7 \pm 0.2$ | $2.0 \pm 0.2$ | - | $2.2 \pm 0.5$ | $2.1 \pm 0.3$ | $2.0 \pm 0.2$ | $2.0 \pm 0.3$ | $2.1 \pm 0.7$ |
| | 168 | $1.4 \pm 0.2$ | $1.7 \pm 0.3$ | $1.9 \pm 0.3$ | $2.8 \pm 0.6$ | - | $2.0 \pm 0.2$ | - | $5.7 \pm 1.2$ | $2.5 \pm 0.6$ |
| rivers | 0 | - | $1.8 \pm 0.3$ | - | - | $2.4 \pm 0.5$ | - | $2.1 \pm 0.0$ | $2.4 \pm 0.4$ | - |
| | 1 | - | $1.8 \pm 0.2$ | - | - | $2.3 \pm 0.5$ | - | $2.1 \pm 0.0$ | $2.2 \pm 0.3$ | - |
| | 3 | - | $1.7 \pm 0.2$ | - | - | $2.4 \pm 0.5$ | - | - | $1.9 \pm 0.4$ | - |
| | 6 | - | $1.6 \pm 0.2$ | - | - | $2.5 \pm 0.6$ | - | $1.9 \pm 0.2$ | $2.2 \pm 0.3$ | $1.8 \pm 0.6$ |
| | 12 | - | $1.5 \pm 0.2$ | - | - | $2.8 \pm 0.7$ | - | $1.6 \pm 0.0$ | - | $1.9 \pm 0.3$ |
| | 24 | - | - | - | - | $2.5 \pm 0.6$ | - | - | $2.1 \pm 0.3$ | $2.2 \pm 0.5$ |
| | 48 | - | - | - | - | $2.0 \pm 0.3$ | - | $2.1 \pm 0.0$ | $2.1 \pm 0.3$ | $2.0 \pm 0.4$ |
| | 72 | - | - | - | - | $1.9 \pm 0.3$ | - | $1.9 \pm 0.2$ | $1.9 \pm 0.3$ | $2.1 \pm 0.4$ |
| | 120 | - | - | - | - | $1.8 \pm 0.4$ | $1.8 \pm 0.0$ | $2.2 \pm 0.3$ | $2.0 \pm 0.0$ | $2.1 \pm 0.4$ |
| | 168 | - | - | - | - | $1.9 \pm 0.4$ | - | $2.1 \pm 0.3$ | $2.0 \pm 0.4$ | $1.7 \pm 0.0$ |
| sst | 0 | $1.3 \pm 0.1$ | - | $1.7 \pm 0.0$ | $1.2 \pm 0.0$ | $2.0 \pm 0.4$ | $2.2 \pm 0.3$ | $1.9 \pm 0.0$ | - | $1.9 \pm 0.4$ |
| | 1 | $1.3 \pm 0.1$ | - | $1.5 \pm 0.0$ | $1.8 \pm 0.5$ | $2.1 \pm 0.5$ | $1.9 \pm 0.3$ | $2.2 \pm 0.0$ | - | $2.1 \pm 0.7$ |
| | 3 | $1.2 \pm 0.2$ | - | - | $1.8 \pm 0.4$ | $2.0 \pm 0.6$ | $2.0 \pm 0.0$ | $1.9 \pm 0.0$ | - | $2.0 \pm 0.7$ |
| | 6 | $1.2 \pm 0.1$ | - | - | $1.4 \pm 0.3$ | $2.2 \pm 0.4$ | $2.1 \pm 0.2$ | $1.7 \pm 0.1$ | - | $2.1 \pm 0.5$ |
| | 12 | $1.3 \pm 0.1$ | - | - | $1.6 \pm 0.0$ | $2.2 \pm 0.1$ | $2.0 \pm 0.2$ | - | $1.6 \pm 0.0$ | $1.9 \pm 0.6$ |
| | 24 | $1.2 \pm 0.1$ | - | - | $1.4 \pm 0.1$ | $1.9 \pm 0.6$ | $2.2 \pm 0.1$ | $1.9 \pm 0.1$ | $2.0 \pm 0.0$ | $1.8 \pm 0.4$ |
| | 48 | $1.2 \pm 0.1$ | - | - | $1.4 \pm 0.0$ | - | - | $1.8 \pm 0.0$ | $1.6 \pm 0.0$ | $2.2 \pm 0.6$ |
| | 72 | - | - | $1.9 \pm 0.0$ | $1.6 \pm 0.3$ | $2.0 \pm 0.4$ | $2.0 \pm 0.0$ | $1.7 \pm 0.2$ | - | $2.4 \pm 0.7$ |
| | 120 | - | - | - | $1.4 \pm 0.0$ | $1.8 \pm 0.2$ | $2.0 \pm 0.3$ | $2.0 \pm 0.1$ | - | $2.4 \pm 0.7$ |
| | 168 | $1.3 \pm 0.0$ | - | - | $1.6 \pm 0.2$ | $2.2 \pm 0.3$ | $2.0 \pm 0.1$ | $2.2 \pm 0.0$ | $1.9 \pm 0.0$ | $2.3 \pm 0.7$ |

**Table A9.** Appendix Table A8 continued.

| Predictor | Delay (h) | Den Helder | Esbjerg | Gedser | Göteborg | Helsinki | Lowestoft | Malmö | Oslo | Umeå |
|---|---|---|---|---|---|---|---|---|---|---|
| $\text{sst}_{\text{ENA}}$ | 0 | - | - | $2.1 \pm 0.0$ | $1.8 \pm 0.3$ | $2.5 \pm 0.6$ | $2.1 \pm 0.2$ | $1.6 \pm 0.0$ | $1.9 \pm 0.3$ | $2.3 \pm 0.5$ |
| | 1 | - | - | $2.0 \pm 0.2$ | $1.7 \pm 0.3$ | $2.4 \pm 0.5$ | $2.1 \pm 0.3$ | - | $1.9 \pm 0.2$ | $2.2 \pm 0.5$ |
| | 3 | $1.3 \pm 0.0$ | - | $2.2 \pm 0.2$ | $1.6 \pm 0.3$ | $2.5 \pm 0.6$ | $2.0 \pm 0.2$ | - | $1.9 \pm 0.3$ | $2.3 \pm 0.5$ |
| | 6 | $1.4 \pm 0.0$ | - | $2.1 \pm 0.1$ | $1.7 \pm 0.4$ | $2.5 \pm 0.5$ | $2.1 \pm 0.1$ | - | $2.1 \pm 0.3$ | $2.3 \pm 0.5$ |
| | 12 | - | - | $1.9 \pm 0.2$ | $1.7 \pm 0.3$ | $2.4 \pm 0.6$ | $1.9 \pm 0.1$ | - | $2.1 \pm 0.2$ | $2.3 \pm 0.4$ |
| | 24 | - | - | $2.0 \pm 0.4$ | $1.7 \pm 0.3$ | $2.4 \pm 0.5$ | $2.0 \pm 0.3$ | - | $1.6 \pm 0.2$ | $2.3 \pm 0.5$ |
| | 48 | $1.3 \pm 0.0$ | - | $1.9 \pm 0.3$ | $1.7 \pm 0.3$ | $2.7 \pm 0.7$ | $2.0 \pm 0.2$ | - | $2.0 \pm 0.3$ | $2.5 \pm 0.6$ |
| | 72 | - | - | $1.8 \pm 0.1$ | $1.9 \pm 0.3$ | $2.5 \pm 0.6$ | $2.1 \pm 0.3$ | - | $1.8 \pm 0.2$ | $2.3 \pm 0.6$ |
| | 120 | - | - | - | $1.7 \pm 0.4$ | $2.4 \pm 0.6$ | $2.0 \pm 0.2$ | $1.8 \pm 0.0$ | $1.8 \pm 0.4$ | $2.1 \pm 0.6$ |
| | 168 | - | - | $2.0 \pm 0.0$ | $1.9 \pm 0.4$ | $2.4 \pm 0.5$ | $2.0 \pm 0.2$ | $1.9 \pm 0.0$ | $2.0 \pm 0.2$ | $1.9 \pm 0.3$ |
| tprecip | 0 | - | $1.6 \pm 0.2$ | $2.2 \pm 0.3$ | $1.5 \pm 0.3$ | - | $2.1 \pm 0.3$ | $1.9 \pm 0.2$ | - | $2.0 \pm 0.5$ |
| | 1 | - | $1.5 \pm 0.2$ | $2.2 \pm 0.3$ | $1.6 \pm 0.3$ | $2.0 \pm 0.5$ | $1.8 \pm 0.2$ | - | - | $2.7 \pm 0.8$ |
| | 3 | $1.3 \pm 0.0$ | - | $2.7 \pm 0.4$ | $1.2 \pm 0.0$ | - | - | $1.8 \pm 0.2$ | - | $4.9 \pm 1.5$ |
| | 6 | - | $1.4 \pm 0.0$ | $2.1 \pm 0.2$ | - | $2.1 \pm 0.4$ | - | - | $1.9 \pm 0.3$ | - |
| | 12 | - | - | $2.3 \pm 0.3$ | $1.7 \pm 0.0$ | $2.3 \pm 0.4$ | - | - | $2.6 \pm 0.5$ | $2.8 \pm 0.8$ |
| | 24 | $1.1 \pm 0.0$ | - | $1.9 \pm 0.2$ | - | $2.0 \pm 0.5$ | - | $2.0 \pm 0.3$ | $1.8 \pm 0.2$ | $1.7 \pm 0.3$ |
| | 48 | - | - | $1.9 \pm 0.3$ | - | $2.0 \pm 0.5$ | $2.0 \pm 0.0$ | $2.3 \pm 0.4$ | $1.8 \pm 0.2$ | $2.1 \pm 0.4$ |
| | 72 | $1.3 \pm 0.1$ | - | - | - | $3.5 \pm 1.1$ | - | $2.2 \pm 0.0$ | $1.7 \pm 0.0$ | $1.9 \pm 0.5$ |
| | 120 | $1.4 \pm 0.2$ | $1.3 \pm 0.0$ | $2.0 \pm 0.1$ | $1.5 \pm 0.2$ | $2.2 \pm 0.5$ | - | - | - | $2.4 \pm 0.6$ |
| | 168 | $1.4 \pm 0.0$ | $1.5 \pm 0.1$ | - | $1.6 \pm 0.3$ | $3.7 \pm 1.0$ | - | $1.9 \pm 0.3$ | - | $2.1 \pm 0.5$ |
| $u_-$ | 0 | - | - | $2.0 \pm 0.2$ | - | - | - | $2.1 \pm 0.3$ | $1.8 \pm 0.2$ | - |
| | 1 | - | - | $1.8 \pm 0.3$ | - | - | - | $1.7 \pm 0.1$ | $1.8 \pm 0.0$ | $2.1 \pm 0.7$ |
| | 3 | - | - | $2.0 \pm 0.3$ | - | - | - | $2.0 \pm 0.3$ | $2.0 \pm 0.4$ | $1.9 \pm 0.3$ |
| | 6 | - | $1.6 \pm 0.2$ | $1.9 \pm 0.2$ | - | - | $2.1 \pm 0.0$ | $2.0 \pm 0.2$ | $1.8 \pm 0.2$ | $1.4 \pm 0.0$ |
| | 12 | - | $1.6 \pm 0.0$ | $1.7 \pm 0.0$ | - | - | - | $2.0 \pm 0.3$ | $2.3 \pm 0.4$ | - |
| | 24 | - | $1.6 \pm 0.2$ | - | - | $2.4 \pm 0.0$ | $2.0 \pm 0.5$ | $2.1 \pm 0.3$ | $2.0 \pm 0.3$ | - |
| | 48 | $1.3 \pm 0.1$ | - | - | - | - | $2.2 \pm 0.2$ | - | $1.8 \pm 0.0$ | $1.8 \pm 0.2$ |
| | 72 | - | - | $1.9 \pm 0.2$ | - | $2.0 \pm 0.0$ | $2.1 \pm 0.2$ | - | - | $1.6 \pm 0.1$ |
| | 120 | $1.3 \pm 0.2$ | - | $2.1 \pm 0.3$ | - | $2.0 \pm 0.2$ | $2.1 \pm 0.3$ | $2.0 \pm 0.5$ | $2.2 \pm 0.4$ | $1.8 \pm 0.5$ |
| | 168 | - | $1.5 \pm 0.0$ | $1.8 \pm 0.1$ | $1.6 \pm 0.3$ | - | $2.0 \pm 0.2$ | $1.9 \pm 0.0$ | $1.5 \pm 0.1$ | $1.4 \pm 0.1$ |

**Table A10.** Appendix Table A8 continued.

| Predictor | Delay (h) | Den Helder | Esbjerg | Gedser | Göteborg | Helsinki | Lowestoft | Malmö | Oslo | Umeå |
|---|---|---|---|---|---|---|---|---|---|---|
| $u_+$ | 0 | $4.6 \pm 0.8$ | $4.2 \pm 0.7$ | $2.1 \pm 0.2$ | $4.6 \pm 1.1$ | $1.9 \pm 0.6$ | $2.6 \pm 0.4$ | $1.8 \pm 0.1$ | $2.5 \pm 0.5$ | - |
| | 1 | $4.3 \pm 0.6$ | $4.2 \pm 0.7$ | $2.0 \pm 0.0$ | $2.3 \pm 0.5$ | $1.9 \pm 0.1$ | $2.6 \pm 0.4$ | - | $2.0 \pm 0.0$ | - |
| | 3 | $4.7 \pm 0.7$ | $3.3 \pm 0.6$ | - | $2.9 \pm 0.8$ | $1.7 \pm 0.0$ | $3.3 \pm 0.6$ | $1.9 \pm 0.3$ | $2.0 \pm 0.0$ | - |
| | 6 | $2.9 \pm 0.5$ | $2.7 \pm 0.5$ | - | $2.5 \pm 0.6$ | $2.6 \pm 0.7$ | $3.6 \pm 0.5$ | $2.1 \pm 0.2$ | $2.2 \pm 0.3$ | $2.2 \pm 0.4$ |
| | 12 | $2.8 \pm 0.5$ | $1.7 \pm 0.2$ | - | $3.4 \pm 0.9$ | - | $6.5 \pm 0.9$ | $4.2 \pm 0.7$ | $2.1 \pm 0.3$ | - |
| | 24 | $1.3 \pm 0.2$ | - | $2.2 \pm 0.3$ | $1.3 \pm 0.0$ | $2.3 \pm 0.5$ | $2.3 \pm 0.3$ | $3.5 \pm 0.7$ | $1.8 \pm 0.3$ | $2.0 \pm 0.5$ |
| | 48 | $1.5 \pm 0.3$ | - | - | $1.8 \pm 0.3$ | $3.8 \pm 1.0$ | $2.1 \pm 0.2$ | $2.2 \pm 0.3$ | $1.9 \pm 0.4$ | $2.3 \pm 0.5$ |
| | 72 | $1.2 \pm 0.1$ | $1.5 \pm 0.1$ | $1.9 \pm 0.3$ | $1.5 \pm 0.2$ | $2.6 \pm 0.6$ | $2.1 \pm 0.3$ | $2.0 \pm 0.2$ | $2.1 \pm 0.4$ | $2.1 \pm 0.9$ |
| | 120 | $1.1 \pm 0.1$ | $1.6 \pm 0.2$ | $2.0 \pm 0.3$ | $1.9 \pm 0.5$ | $2.3 \pm 0.6$ | $2.0 \pm 0.0$ | $2.5 \pm 0.4$ | - | $2.1 \pm 0.5$ |
| | 168 | $1.4 \pm 0.0$ | $1.5 \pm 0.1$ | $2.0 \pm 0.3$ | $1.5 \pm 0.2$ | $3.5 \pm 1.2$ | $2.1 \pm 0.2$ | $2.3 \pm 0.3$ | - | $2.6 \pm 0.6$ |
| $v_-$ | 0 | $1.5 \pm 0.2$ | - | $3.4 \pm 0.6$ | - | $2.1 \pm 0.4$ | $2.2 \pm 0.2$ | $2.5 \pm 0.5$ | - | - |
| | 1 | $1.5 \pm 0.2$ | - | $3.7 \pm 0.7$ | - | $2.0 \pm 0.4$ | $2.0 \pm 0.3$ | $3.0 \pm 0.6$ | - | - |
| | 3 | $1.7 \pm 0.2$ | - | $4.6 \pm 0.8$ | - | $2.1 \pm 0.1$ | $2.4 \pm 0.3$ | $4.6 \pm 0.9$ | - | - |
| | 6 | $1.9 \pm 0.3$ | - | $5.0 \pm 1.0$ | - | $1.8 \pm 0.5$ | $2.0 \pm 0.2$ | $6.2 \pm 1.1$ | - | - |
| | 12 | $1.5 \pm 0.0$ | - | $4.1 \pm 0.8$ | - | - | $2.1 \pm 0.1$ | $3.3 \pm 0.7$ | - | - |
| | 24 | - | $1.6 \pm 0.0$ | $2.0 \pm 0.2$ | - | $2.0 \pm 0.4$ | $2.2 \pm 0.1$ | $2.5 \pm 0.4$ | - | - |
| | 48 | $1.2 \pm 0.1$ | - | $2.3 \pm 0.4$ | - | - | $2.0 \pm 0.2$ | $2.0 \pm 0.3$ | - | - |
| | 72 | - | - | $2.0 \pm 0.1$ | - | - | - | - | $1.8 \pm 0.0$ | - |
| | 120 | $1.4 \pm 0.2$ | $1.6 \pm 0.2$ | $1.7 \pm 0.0$ | - | - | - | $2.1 \pm 0.3$ | $1.9 \pm 0.3$ | - |
| | 168 | $1.2 \pm 0.0$ | - | $1.9 \pm 0.2$ | - | - | $2.1 \pm 0.1$ | $1.4 \pm 0.0$ | - | - |
| $v_+$ | 0 | $1.3 \pm 0.2$ | $1.4 \pm 0.2$ | - | $1.5 \pm 0.2$ | $1.8 \pm 0.6$ | - | - | $2.1 \pm 0.3$ | $2.1 \pm 0.5$ |
| | 1 | $1.3 \pm 0.2$ | $1.4 \pm 0.0$ | $2.3 \pm 0.0$ | $1.8 \pm 0.3$ | $2.0 \pm 0.0$ | - | - | $2.0 \pm 0.3$ | $2.1 \pm 0.4$ |
| | 3 | $1.2 \pm 0.4$ | $1.4 \pm 0.1$ | $1.9 \pm 0.3$ | $1.3 \pm 0.4$ | - | $1.7 \pm 0.1$ | - | $2.0 \pm 0.2$ | $2.3 \pm 0.6$ |
| | 6 | $1.5 \pm 0.2$ | $1.5 \pm 0.1$ | $2.0 \pm 0.2$ | - | $2.1 \pm 0.2$ | $1.8 \pm 0.0$ | - | $1.7 \pm 0.2$ | $2.6 \pm 0.7$ |
| | 12 | - | $1.4 \pm 0.2$ | - | $1.6 \pm 0.3$ | $2.1 \pm 0.4$ | - | $1.9 \pm 0.2$ | $2.2 \pm 0.3$ | $4.4 \pm 1.5$ |
| | 24 | $1.3 \pm 0.1$ | $2.4 \pm 0.3$ | $2.1 \pm 0.3$ | $1.5 \pm 0.2$ | $3.1 \pm 0.8$ | - | $2.3 \pm 0.4$ | $2.4 \pm 0.5$ | $6.1 \pm 1.6$ |
| | 48 | - | - | $1.8 \pm 0.0$ | - | $2.3 \pm 0.6$ | - | - | - | - |
| | 72 | - | $1.5 \pm 0.1$ | $2.0 \pm 0.1$ | $1.5 \pm 0.2$ | $1.9 \pm 0.3$ | - | $2.0 \pm 0.2$ | $2.0 \pm 0.3$ | $2.3 \pm 0.5$ |
| | 120 | $1.2 \pm 0.1$ | $1.5 \pm 0.3$ | - | $1.6 \pm 0.3$ | - | - | - | $2.0 \pm 0.3$ | $2.2 \pm 0.0$ |
| | 168 | - | - | $2.1 \pm 0.3$ | $2.8 \pm 0.8$ | $2.4 \pm 0.6$ | - | $2.0 \pm 0.3$ | $2.2 \pm 0.4$ | - |

**Table A11.** Appendix Table A8 continued.

| Predictor | Delay (h) | Den Helder | Esbjerg | Gedser | Göteborg | Helsinki | Lowestoft | Malmö | Oslo | Umeå |
|-----------|-----------|------------|---------|--------|----------|----------|-----------|-------|------|------|
| wdir | 0 | $1.4 \pm 0.2$ | $1.2 \pm 0.0$ | $2.1 \pm 0.3$ | - | $2.4 \pm 0.8$ | - | $2.8 \pm 0.5$ | $2.0 \pm 0.4$ | - |
| | 1 | $1.5 \pm 0.2$ | - | $1.9 \pm 0.2$ | - | $2.6 \pm 0.6$ | - | $2.3 \pm 0.4$ | - | - |
| | 3 | $1.5 \pm 0.2$ | - | $1.9 \pm 0.3$ | - | $2.2 \pm 0.4$ | $2.0 \pm 0.2$ | $2.1 \pm 0.3$ | - | - |
| | 6 | $1.8 \pm 0.2$ | $1.4 \pm 0.0$ | $3.2 \pm 0.6$ | - | $2.0 \pm 0.4$ | $2.2 \pm 0.3$ | $2.7 \pm 0.5$ | $1.8 \pm 0.0$ | $1.8 \pm 0.2$ |
| | 12 | $1.1 \pm 0.0$ | $1.4 \pm 0.1$ | $2.2 \pm 0.4$ | $1.5 \pm 0.0$ | $1.9 \pm 0.0$ | $1.8 \pm 0.3$ | $2.2 \pm 0.3$ | $1.9 \pm 0.3$ | - |
| | 24 | $1.3 \pm 0.0$ | $1.6 \pm 0.1$ | $1.9 \pm 0.2$ | - | $3.3 \pm 0.9$ | - | $2.2 \pm 0.1$ | - | $2.5 \pm 0.0$ |
| | 48 | - | - | $1.8 \pm 0.2$ | - | $2.2 \pm 0.4$ | $2.2 \pm 0.2$ | $2.0 \pm 0.2$ | - | $2.0 \pm 0.4$ |
| | 72 | - | - | $2.4 \pm 0.4$ | - | $2.0 \pm 0.4$ | - | $2.0 \pm 0.0$ | - | $2.2 \pm 0.5$ |
| | 120 | - | - | $2.4 \pm 0.4$ | - | - | $1.9 \pm 0.0$ | $2.1 \pm 0.4$ | $1.5 \pm 0.0$ | - |
| | 168 | $1.2 \pm 0.1$ | - | $2.0 \pm 0.2$ | $1.5 \pm 0.2$ | $2.0 \pm 0.2$ | - | $2.2 \pm 0.3$ | - | - |
| wspeed | 0 | $9.5 \pm 1.2$ | $6.9 \pm 0.9$ | $3.2 \pm 0.6$ | $8.3 \pm 1.4$ | $2.1 \pm 0.5$ | $4.3 \pm 0.6$ | $2.0 \pm 0.2$ | $2.2 \pm 0.4$ | $1.7 \pm 0.2$ |
| | 1 | $9.0 \pm 1.1$ | $5.7 \pm 0.8$ | $4.7 \pm 0.8$ | $7.8 \pm 1.4$ | $1.9 \pm 0.0$ | $4.1 \pm 0.6$ | $1.6 \pm 0.3$ | $2.2 \pm 0.4$ | $1.4 \pm 0.0$ |
| | 3 | $11.1 \pm 1.3$ | $6.2 \pm 0.8$ | $6.0 \pm 1.0$ | $5.9 \pm 1.2$ | $2.0 \pm 0.5$ | $5.8 \pm 0.8$ | $1.9 \pm 0.3$ | $2.8 \pm 0.5$ | $2.1 \pm 0.5$ |
| | 6 | $8.6 \pm 1.1$ | $4.5 \pm 0.8$ | $3.0 \pm 0.6$ | $3.6 \pm 0.9$ | $2.4 \pm 0.5$ | $3.9 \pm 0.6$ | $4.1 \pm 0.8$ | $2.2 \pm 0.4$ | $2.3 \pm 0.7$ |
| | 12 | $4.0 \pm 0.6$ | $2.4 \pm 0.4$ | $3.1 \pm 0.6$ | $3.0 \pm 0.8$ | $2.3 \pm 0.5$ | $3.2 \pm 0.5$ | $6.2 \pm 1.0$ | - | $4.6 \pm 0.9$ |
| | 24 | $1.5 \pm 0.2$ | $2.1 \pm 0.3$ | $2.4 \pm 0.3$ | $2.0 \pm 0.4$ | $2.4 \pm 0.6$ | $1.9 \pm 0.1$ | $4.1 \pm 0.9$ | $2.7 \pm 0.6$ | $6.2 \pm 1.8$ |
| | 48 | $1.7 \pm 0.3$ | $1.5 \pm 0.1$ | $2.0 \pm 0.3$ | $1.6 \pm 0.3$ | $2.6 \pm 0.7$ | $2.5 \pm 0.4$ | - | - | $2.0 \pm 0.4$ |
| | 72 | $1.3 \pm 0.2$ | $1.5 \pm 0.1$ | $2.1 \pm 0.2$ | $1.8 \pm 0.3$ | $2.1 \pm 0.3$ | $1.8 \pm 0.1$ | $1.8 \pm 0.2$ | $2.6 \pm 0.5$ | $2.2 \pm 0.6$ |
| | 120 | - | $1.7 \pm 0.2$ | $2.1 \pm 0.3$ | $2.1 \pm 0.7$ | $2.0 \pm 0.4$ | $1.9 \pm 0.2$ | $2.4 \pm 0.4$ | $1.6 \pm 0.0$ | $2.1 \pm 0.5$ |
| | 168 | $1.3 \pm 0.1$ | $1.5 \pm 0.2$ | $2.1 \pm 0.2$ | $2.0 \pm 0.5$ | $2.7 \pm 0.6$ | $1.6 \pm 0.0$ | $2.6 \pm 0.4$ | $1.4 \pm 0.0$ | - |

**Table A12.** Appendix Table A8 continued.

*Author contributions.* CH and HR came up with the original idea and funding for the project. The idea was further developed and preliminary tests conducted by LC under the supervision of LP. CH led the presented work, with input and software from LC and HR, and data from LP. CH wrote the original draft. All authors revised the manuscript.

*Competing interests.* The authors declare that they have no conflict of interest.

*Acknowledgements.* This work is funded by FORMAS grant 2020-00982 awarded to CH. We thank the two anonymous reviewers for their comments that increased the clarity of this manuscript.

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
