# Peer review of "Drivers of high frequency extreme sea level around Northern Europe - Synergies between recurrent neural networks and Random Forest"

_EGUsphere, 2025_

## Author Response (AR1)

All reviewer comments have been addressed, as described below, starting with Reviewer 1 (pages 1 – 4) then Reviewer 2 (pages 5-6). The reviewer comments are indicated with **bold fonts**; our responses, with plain fonts.

The contribution of the two anonymous reviewers has been added to the acknowledgement section.

**Reviewer 1**

**The paper investigates the impact of different physical influences on extreme sea levels in coastal Northern Europe by replacing the values of these physical fields with random noise during training with LSTMs and random forests. These two architectures were chosen because they offer some explainability and interpretation. The study is sound and properly done, the plots are clear - but it largely confirms already known drivers from numerical modeling. I suggest the authors better illuminate for the readers what novelty they are bringing with this particular paper. I think the paper would benefit from addressing the comments below before i can recommend publication.**

We thank the reviewer for their comments. Note that the contribution of the reviewer was duly acknowledged in the acknowledgement section.

The fact that the real-world-data-fed machine learning results confirm the relationships suspected by numerical modelling is to us our most important result: as great and useful as numerical models are, they are only models and can have very large biases, including unrealistic processes (see e.g. the many publications by this team on ocean models). We have rephrased the paper throughout to make this message clearer, and have addressed your other comments as detailed below.

**L44: the cited HIDRA2 paper does indeed note that the model has problems reproducing high frequency variability, but as far as i can tell it mentions no problem with extreme values. Please check.**

They mention throughout section 3.2 that their results are too "smooth", and their figure 6 shows that the most extreme values are underestimated. You are however correct that this is not what they wrote. We therefore rephrased our text.

**L57: do I understand correctly that past (eg. one day before the forecast) tide gauge observations are not used as input? Do you think the network cannot learn anything from past sea levels?**

We considered including past sea level in our network, as it is common in such applications. However, our primary objective is not sea level prediction per se, but the determination of the drivers of extreme events. Including past sea level as a potential predictor would not have taught us anything – we know that there is a large autocorrelation in the sea level signal, so we decided not to include it.

**L83: are MSL_ice and MSL_azo spatial means or spatiotemporal means over the respective domain? Perhaps i missed this...How exactly do you generate the remote drivers? Please explain in a bit more detail.**

We rephrased to clarify that these are hourly timeseries, and specify in each bullet point "hourly spatial-minimum", "hourly spatial-maximum", and "hourly spatial-mean".

**L99: you write that you opted to choose only residuals because the tides are deterministic. While that is a legitimate choice, networks can learn from data that we often find redundant. However, since the Baltic is a low tide basin, your decision might make sense. The only way to know for sure would be to repeat your analysis with the tides present but i don't expect it would make much difference. Perhaps it would indeed worsen the results, especially using such a simple method as a random forest.**

You are right that this could be an interesting experiment, since as you say the Baltic has virtually no tide but the North Sea has some of the strongest in the world. However, given that the LSTM training takes a lot of computing resources, and that de-tiding the timeseries is a common procedure (see literature cited in our manuscript), we shall leave this experiment to others.

**L142: what was the space for hyperparameter search? Which hyperparameters were modified within which intervals?**

We added the lists to the LSTM description in section 2.4.1 and to the Random Forest description in section 2.4.2.

**L246: I am not sure I agree with the formulation that precipitation is an important driver of the sea levels. It may be an important predictor and the net can learn something from it, but the driver indicates a dynamic relationship. So precipitation may perhaps be a proxy for low air pressure which is the actual driver, but is not the driver itself. Please pay attention to this in the text.**

We politely disagree. Precipitation directly adds water on top of the ocean surface, water which is fresh and will hence have a steric contribution. Besides, since the grid cell is rather large it also includes precipitation on land, which is helpful in our case since the river runoff timeseries is low temporal resolution.

However, this is not what we meant by "precipitation is important" L246. We meant that the network finds a high importance for precipitation. We rephrased to make this clearer.

**L255-262: You can probably test the impact of global warming by limiting testing data first to, for example, year 1998, and then to year 2024. The signal of global warming should lead to some systematic differences in these two years...**

Internal variability is large in the climate system, so testing on individual years would not be sufficient. For example, 1998 that I assume you chose randomly had a strong El Niño. Another example was performed by our co-author Linn Carlstedt, who compared years with positive and negative NAO in her Master's thesis: https://gupea.ub.gu.se/bitstream/handle/2077/76416/B1218.pdf.

To test this properly, one would need two distinct 30-year periods, which is not possible for all our locations. This is however an interesting suggestion for a future study that we shall bear in mind, thank you.

**L279: you state that your method would remain functioning under SLR scenarios because you work with residuals, not the full SSH. I am not sure I understand why any method using full SSH would fail to work under any SLR scenario. Tides are also an anomaly of the MSL so the drift in the MSL does not affect the tides (in the linear regime at least). Just as any full SSH network might need to be retrained under future sea levels, so your network would need to be retrained under future climate. I see no clear benefit to the residual approach in this regard...I would suggest to rephrase this.**

I am not sure I understand your comment. Tides are already changing in response to sea level rise, since the bottom depth and basin geometry are modified, which also impact wave propagation (e.g. Flick et al. 2003; Woodworth 2010; Schindelegger et al. 2018), and are projected to carry on changing under the different warming scenarios (e.g. for a Northern Europe focus Pickering et al. 2012 or Idier 2017).

And no, we argue that we do not need to retrain our model as long as the drivers of sea level extremes remain the same, i.e. as long as there is no tipping point changing which driver is most important. In fact, this is what we are doing for the rest of this project: we retrained the network to work with historical climate model data, and now use it to predict changes in future sea level extremes.

We added a short text and references to summarise this.

**L285-286: River runoff is not irrelevant for sea levels because it would not be collocated or would have only daily means of measurements, as you state. River runoff is simply not physically anywhere nearly as important as pressure and winds. A river will never give you a basin scale surge of 1m and it will never establish a huge sea level gradient because any SSH gradient due to the river runoff would immediately be radiated away by the gravity waves. Please rephrase this paragraph.**

This is not what we wrote. We wrote that "our models found that rivers are not important for extreme sea level" L286, and throughout the manuscript explained why this result was 1. surprising and 2. probably incorrect, caused by the low quality of the runoff timeseries. We added an extra "surprisingly and contrary to common knowledge" before this sentence to make sure that this point is as clear as possible.

**L300: The Barzandeh paper that you cite does not seem to use LSTMs, it's a CNN architecture. Please check this.**

You are correct. According to Rus et al. (2023), HIDRA2 does not feature any convLSTM layer as we had incorrectly read but a series of conv2D. We rephrased.

**Reviewer 2**

**The paper illustrates the impact of incorporating random noise into sea-level simulations in the North Atlantic, employing LSTM and random forest approaches. The manuscript is structured and discussed in a homogeneous and detailed manner and is practically ready for publication, aside from minor revisions.**

We thank the reviewer for their comments. Note that the contribution of the reviewer was duly acknowledged in the acknowledgement section. We have addressed all your comments, as detailed below.

**-How can precipitation serve as a predictor of sea-level rise? Although this can be reasonably inferred, it is important to explicitly explain this relationship within the manuscript.**

See similar comment from Reviewer 1. Precipitation directly adds water on top of the ocean surface, water which is fresh and will hence have a steric contribution. Besides, since the grid cell is rather large it also includes precipitation on land, which is helpful in our case since the river runoff timeseries is low temporal resolution. We added a short explanation to the text, for all predictors.

**-Could the method be applied in other regions? If so, what test areas might be used to evaluate its broader applicability, strengths, and weaknesses? Consider adding a few lines addressing this in the conclusion.**

The model can probably be used with minimum re-training on other mid-latitude regions. At high latitudes, the absence of sea ice will be problematic. At low latitudes, other remote drivers may be more relevant to include, in particular those associated with cyclone activity, but this is beyond our area of expertise. We added some text in the conclusion as suggested.

**-How might this model perform under climate change scenarios involving large-scale shifts in synoptic patterns? Would the approach remain applicable? If so, this argument should be strengthened.**

See similar comment on climate change from Reviewer 1. We added several sentences to section 3.3 addressing future climate change. To answer your question in particular, we expect that the main impact would be having to change the definition of the remote drivers that correspond to the storm track.

**-Briefly discuss the potential impact of omitted predictors, such as ice cover, on your results.**

The text already mentioned: "There are also observations that we wish we could have included but, to the best of our knowledge, do not exist over the long time period needed for the training, such as the sea ice for Umeå. This could explain why our predictions never explained all the variance in the time series."

We have added the extra, brief explanation: "However, since all our LSTM predictions explained more than two thirds of the signal, we are confident that we included the main predictors."

---

## Referee Report (RR1)

I am generally satisfied with the response and the revision of the paper and I recommend publication. I wasn't clear obviously in some of my comments but i will leave it as it is since my misunderstood comments are not of such importance.

I have only one explanation I would like to make. Below is my objection to the claim that precipitation is an important driver of the sea level and the manuscript authors' response.

**Reviewer comment: L246: I am not sure I agree with the formulation that precipitation is an important driver of the sea levels. It may be an important predictor and the net can learn something from it, but the driver indicates a dynamic relationship. So precipitation may perhaps be a proxy for low air pressure which is the actual driver, but is not the driver itself. Please pay attention to this in the text.**

**Author response: We politely disagree. Precipitation directly adds water on top of the ocean surface, water which is fresh and will hence have a steric contribution. Besides, since the grid cell is rather large it also includes precipitation on land, which is helpful in our case since the river runoff timeseries is low temporal resolution. However, this is not what we meant by "precipitation is important" L246. We meant that the network finds a high importance for precipitation. We rephrased to make this clearer.**

I think I need to clarify my response. I am not sure what the authors have in mind about how a steric contribution looks like but I'll try to explain what I mean by my comment. It is true that freshwater does add to the ocean surface as a steric contribution. However, let's take a look at an apocalyptic storm with a 100 mm/hour rainfall rate. If a model grid cell would be sealed off from the rest of the ocean, this storm would add 10 cm to the sea level in the grid cell in one hour. This of course would never happen in nature because the grid cell is not isolated from the neighbouring ocean. Momentum conservation, combined with the continuity equation, indicates that any sea level gradient induced by the rainfall gets immediately radiated away in the form of surface gravity waves. And these waves travel fast with barotropic speeds (because pressure gradient force is depth independent) and these speeds are much faster than 10 cm/hour which is the accumulation rate of an intense storm. So 100 mm/hour rainfall would never, in an open basin, add 10 cm to the sea level. My point is that there is a steric contribution, but it is in the form of gravity waves which rapidly radiate into infinity and reduce sea level gradient due to precipitation. In this sense precipitation is not a driver of storm surges. It just coincidentally occurs with pressure lows and wind stresses which are the actual drivers of far more importance than adding rainfall on top of an ocean.